# AUTOREGRESSIVE DIFFUSION MODELS

**Emiel Hoogeboom,**[*] **Alexey A. Gritsenko, Jasmijn Bastings, Ben Poole,**
**Rianne van den Berg, Tim Salimans**
Google Research
e.hoogeboom@uva.nl,{agritsenko,pooleb,bastings,salimans}@google.com,
riannevdberg@gmail.com

## ABSTRACT

We introduce Autoregressive Diffusion Models (ARDMs), a model class encompassing and generalizing order-agnostic autoregressive models (Uria et al., 2014) and absorbing discrete diffusion (Austin et al., 2021), which we show are special cases of ARDMs under mild assumptions. ARDMs are simple to implement and easy to train. Unlike standard ARMs, they do not require causal masking of model representations, and can be trained using an efficient objective similar to modern probabilistic diffusion models that scales favourably to highly-dimensional data. At test time, ARDMs support parallel generation which can be adapted to fit any given generation budget. We find that ARDMs require significantly fewer steps than discrete diffusion models to attain the same performance. Finally, we apply ARDMs to lossless compression, and show that they are uniquely suited to this task. Contrary to existing approaches based on bits-back coding, ARDMs obtain compelling results not only on complete datasets, but also on compressing single data points. Moreover, this can be done using a modest number of network calls for (de)compression due to the model's adaptable parallel generation.

## 1 INTRODUCTION

Deep generative models have made great progress in modelling different sources of data, such as images, text and audio. These models have a wide variety of applications, such as denoising, inpainting, translating and representation learning. A popular type of likelihood-based models are Autoregressive Models (ARMs). ARMs model a high-dimensional joint distribution as a factorization of conditionals using the probability chain rule. Although very effective, ARMs require a pre-specified order in which to generate data, which may not be an obvious choice for some data modalities, for example images. Further, although the likelihood of ARMs can be retrieved with a single neural network call, sampling from a model requires the same number of network calls as the dimensionality of the data.

Recently, modern probabilistic diffusion models have introduced a new training paradigm: Instead of optimizing the entire likelihood of a datapoint, a component of the likelihood bound can be sampled and optimized instead. Works on diffusion on discrete spaces (Sohl-Dickstein et al., 2015;

---

[*]Work done during as research intern at Google Brain.

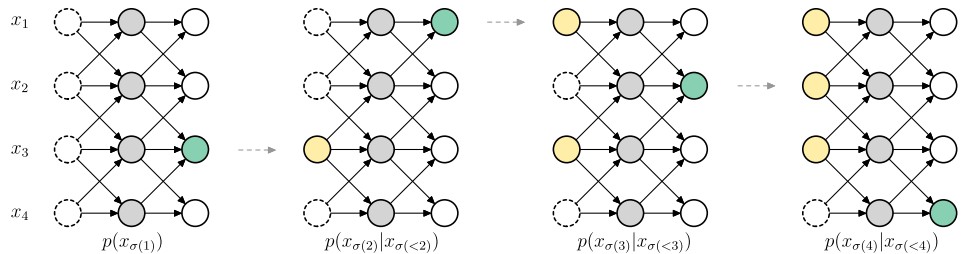

$$p(x_{\sigma(1)}) \qquad p(x_{\sigma(2)}|x_{\sigma(<2)}) \qquad p(x_{\sigma(3)}|x_{\sigma(<3)}) \qquad p(x_{\sigma(4)}|x_{\sigma(<4)})$$

Figure 1: Generation of Autoregressive Diffusion Models for the generation order $\sigma = (3, 1, 2, 4)$. Filled circles in the first and third layers represent respectively the input and output variables, and the middle layer represents internal activations of the network.

Hoogeboom et al., 2021; Austin et al., 2021) describe a discrete destruction process for which the inverse generative process is learned with categorical distributions. However, the length of these processes may need to be large to attain good performance, which leads to a large number of network calls to sample from or evaluate the likelihood with discrete diffusion.

In this work we introduce Autoregressive Diffusion Models (ARDMs), a variant of autoregressive models that learns to generate in any order. ARDMs generalize order agnostic autoregressive models and discrete diffusion models. We show that ARDMs have several benefits: In contrast to standard ARMs, they impose no architectural constraints on the neural networks used to predict the distribution parameters. Further, ARDMs require significantly fewer steps than absorbing models to attain the same performance. In addition, using dynamic programming approaches developed for diffusion models, ARDMs can be parallelized to generate multiple tokens simultaneously without a substantial reduction in performance. Empirically we demonstrate that ARDMs perform similarly to or better than discrete diffusion models while being more efficient in modelling steps. The main contributions of this paper can be summarized as follows: 1) We introduce ARDMs, a variant of order-agnostic ARMs which include the ability to upscale variables. 2) We derive an equivalence between ARDMs and absorbing diffusion under a continuous time limit. 3) We show that ARDMs can have parallelized inference and generation processes, a property that among other things admits competitive lossless compression with a modest number of network calls.

## 2 BACKGROUND

ARMs factorize a multivariate distribution into a product of $D$ univariate distributions using the probability chain rule. In this case the log-likelihood of such as model is given by:

$$\log p(\boldsymbol{x}) = \sum_{t=1}^{D} \log p(x_t | \boldsymbol{x}_{<t}), \tag{1}$$

where $\boldsymbol{x}_{<t}$ is shorthand for $x_1, x_2, \ldots, x_{t-1}$. ARMs are trained by ensuring that the neural network has a triangular dependency structure, for instance implemented via causal masking. Although this allows parallelized computation of the likelihood for all conditional distributions at once, to sample from the model it requires $D$ iterative sampling steps $x_1 \sim p(x_1), x_2 \sim p(x_2|x_1)$ towards $x_D \sim p(x_D | x_1, x_2, \ldots, x_{D-1})$.

**Order Agnostic ARMs** Order Agnostic ARMs (OA-ARMs) (Uria et al., 2014) generate variables with a random ordering $\sigma \in S_D$, where $S_D$ represents a set of all permutations of the integers $1, \ldots, D$. The log-likelihood of this model is given by:

$$\log p(\boldsymbol{x}) \geq \mathbb{E}_{\sigma \sim \mathcal{U}(S_D)} \sum_{t=1}^{D} \log p(x_{\sigma(t)} | \boldsymbol{x}_{\sigma(<t)}). \tag{2}$$

This can be seen as a latent variable model and the log-likelihood is derived via Jensen's inequality:

$$\log p(\boldsymbol{x}) = \log \mathbb{E}_{\sigma \sim \mathcal{U}(S_D)} p(\boldsymbol{x}|\sigma) \geq \mathbb{E}_{\sigma \sim \mathcal{U}(S_D)} \log p(\boldsymbol{x}|\sigma).$$

Mind that hereafter we leave out $\sigma$ in our notation to avoid clutter. One approach to train OA-ARMs is the procedure as described by Yang et al. (2019) for XLNet. It takes a permutation equivariant network such as a Transformer that is causally masked. Then, inputs are permuted and outputs are permuted back according to a given order, which models the sequence in that specific order. However, such approaches typically suffer in likelihood score and cannot be combined with simple non-equivariant transformations such as convolutional layers.

**Discrete Diffusion** Discrete diffusion models define a destruction process on discrete data. An example is absorbing diffusion (Austin et al., 2021), for which each variable has a probability of decaying to an absorbing state. The opposite process to the destruction process is the learned *generative process*. This generative process models the distribution over variables that are currently absorbed, and generates these with a probability.

## 3 AUTOREGRESSIVE DIFFUSION MODELS

We introduce Autoregressive Diffusion Models (ARDMs). ARDMs generate variables in an *arbitrary order*, one by one. Further, ARDMs are able to *upscale* variables, such as the bit values of a

| **Algorithm 1** Sampling from OA-ARDMs | **Algorithm 2** Optimizing OA-ARDMs |
|---|---|
| **Input:** Network $f$ | **Input:** Datapoint $\boldsymbol{x}$, Network $f$ |
| **Output:** Sample $\boldsymbol{x}$ | **Output:** ELBO $\mathcal{L}$ |
| Initialize $\boldsymbol{x} = \boldsymbol{0}$ | Sample $t \sim \mathcal{U}(1, \ldots, D)$ |
| Sample $\sigma \sim \mathcal{U}(S_D)$ | Sample $\sigma \sim \mathcal{U}(S_D)$ |
| **for** $t$ in $\{1, \ldots, D\}$ **do** | Compute $\boldsymbol{m} \leftarrow (\sigma < t)$ |
| $\quad \boldsymbol{m} \leftarrow (\sigma < t)$ and $\boldsymbol{n} \leftarrow (\sigma = t)$ | $\boldsymbol{l} \leftarrow (1 - \boldsymbol{m}) \odot \log \mathcal{C}(\boldsymbol{x}\|f(\boldsymbol{m} \odot \boldsymbol{x}))$ |
| $\quad \boldsymbol{x}' \sim \mathcal{C}(\boldsymbol{x}\|f(\boldsymbol{m} \odot \boldsymbol{x}))$ | $\mathcal{L}_t \leftarrow \frac{1}{D-t+1} \mathrm{sum}(\boldsymbol{l})$ |
| $\quad \boldsymbol{x} \leftarrow (1 - \boldsymbol{n}) \odot \boldsymbol{x} + \boldsymbol{n} \odot \boldsymbol{x}'$ | $\mathcal{L} \leftarrow D \cdot \mathcal{L}_t$ |

pixel. Unlike standard ARMs, ARDMs are trained on a *single* step in the objective, as in modern diffusion models. In addition, both sampling and inference of ARDMs can be parallelized using dynamic programming with minimal degradation in log-likelihood.

**Order Agnostic ARDMs** The main difficulty of parameterizing an autoregressive model from an engineering perspective, is the need to enforce the triangular or causal dependence. Especially for 2D signals, this triangular dependence is difficult to enforce for arbitrary orders (Jain et al., 2020) and tedious design is needed for multi-scale architectures (Salimans et al., 2017). To relax this requirement, we take inspiration from modern diffusion-based generative models. Using these insights, we derive an objective that is only optimized for a single step at a time. Starting at Equation 2, a different objective for an order agnostic ARM can be derived, by replacing the summation over $t$ by an expectation that is appropriately re-weighted:

$$
\begin{aligned}
\log p(\boldsymbol{x}) &\geq \mathbb{E}_{\sigma \sim \mathcal{U}(S_D)} \sum_{t=1}^{D} \log p(x_{\sigma(t)}|\boldsymbol{x}_{\sigma(<t)}) \\
&= \mathbb{E}_{\sigma \sim \mathcal{U}(S_D)} D \cdot \mathbb{E}_{t \sim \mathcal{U}(1,\ldots,D)} \log p(x_{\sigma(t)}|\boldsymbol{x}_{\sigma(<t)}) \\
&= D \cdot \mathbb{E}_{t \sim \mathcal{U}(1,\ldots,D)} \mathbb{E}_{\sigma \sim \mathcal{U}(S_D)} \frac{1}{D-t+1} \sum_{k \in \sigma(\geq t)} \log p(x_k|\boldsymbol{x}_{\sigma(<t)})
\end{aligned}
$$

Compactly, we can write the expected lower bound as:

$$
\log p(\boldsymbol{x}) \geq \mathbb{E}_{t \sim \mathcal{U}(1,\ldots,D)}[D \cdot \mathcal{L}_t], \text{ where } \mathcal{L}_t = \frac{1}{D-t+1} \mathbb{E}_{\sigma \sim \mathcal{U}(S_D)} \sum_{k \in \sigma(\geq t)} \log p(x_k|\boldsymbol{x}_{\sigma(<t)}).
$$

Here the term $\mathcal{L}_t$ represents the likelihood component for step $t$. Importantly, we do not need to optimize for all $\mathcal{L}_t$ terms of a datapoint simultaneously. Instead, for each datapoint in a minibatch a single $\mathcal{L}_t$ term is optimized where $t$ is sampled from a uniform distribution. This objective was originally proposed by Uria et al. (2014) to train order-agnostic ARMs. We will develop ARDMs starting from this perspective and refer to the special case of an order-agnostic ARM, as an order agnostic ARDM (OA-ARDM). Interestingly, each $\mathcal{L}_t$ component can be seen as a BERT-like training objective (Devlin et al., 2019), where exactly $D - t + 1$ tokens are masked and subsequently predicted. Therefore, an OA-ARDM is trained as a collection of $D$ BERTs with loss terms $\mathcal{L}_t$, which contain the reweighting term $\frac{1}{D-t+1}$. Another insight is that this generative process is very similar to absorbing diffusion, where the model aims to generate absorbed (or masked) variables. In certain situations we might want to refer to loss terms instead of likelihood terms, so we define $L_t = -\mathcal{L}_t$.

**Parametrization** We desire a parametrization for the model distribution $\log p(x_k|\boldsymbol{x}_{\sigma(<t)})$ for $k \in \sigma(\geq t)$ for all $\sigma$ and $t$. For each $\sigma$ and $t$ it is in principle allowed to have an entirely new neural network. However, this would be very inconvenient as the number of $t$ grows as $\mathcal{O}(D)$ and the number of $\sigma$ grows as $\mathcal{O}(D!)$. Instead, a single neural network is utilized and shared for different $\sigma$ and $t$. This is implemented by masking variables at the input, and predicting those at the output. To be precise, we let $\boldsymbol{x} \in \mathcal{X} = \{1, 2, \ldots, K\}^D$ represent discrete variables with $K$ classes and a neural network $f : \mathcal{X} \rightarrow \mathbb{R}^{D \times K}$ that outputs probability vectors for each dimension. Conditioning is done via masking: For a given permutation array $\sigma$, we compute the element-wise comparison $\boldsymbol{m} = \sigma < t$ which produces a Boolean

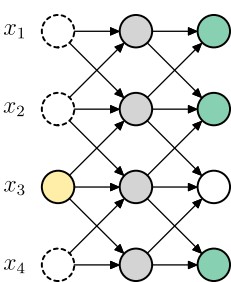

Figure 2: ARDM training step. This step optimizes for step $t = 2$ for all possible permutations $\sigma$ simultaneously which satisfy $\sigma(1) = 3$.

mask. The mask is then used by predicting $\boldsymbol{\theta} = f(\boldsymbol{m} \odot \boldsymbol{x})$, where $\odot$ denotes element-wise multiplication. For each location $k \in \sigma(\geq t)$, the log probability vectors $\boldsymbol{\theta}_k$ are used. Letting $\mathcal{C}(x_k|\boldsymbol{\theta}_k)$ denote a categorical distribution over $x_k$ with class probabilities $\boldsymbol{\theta}_k$, we choose to model $\log(x_k|\boldsymbol{x}_{\sigma(<t)}) = \log \mathcal{C}(x_k|\boldsymbol{\theta}_k)$. The locations of these relevant indices $k \in \sigma(\geq t)$ are retrieved by using the opposite mask $\boldsymbol{1} - \boldsymbol{m}$. The procedure to sample and optimize an ARDM with this parametrization are given in Algorithms 1 and 2. A training step is visualized in Figure 2. Note that opposed to Figure 1 where only a single output was used per step, in the train step all dimensions that were masked are predicted simultaneously. Therefore, there are multiple variables to predict which ensures there is sufficient signal to optimize the model.

For clarity some of the implementation details have been left out in the explanation above. However, given that these are important for practical implementations, they are specified in the following. The input to the function $f$ may be different depending on the data modality: For images and audio, the mask is applied to the input so that values are zero *after* feature normalization. The mask itself is also concatenated to the input as an input representation, which allows the model to identify whether a value is actually zero, or the value is in the absorbing state zero. For language the input representation is augmented and absorbed values are instead set to a new class $K + 1$, in which case there is no need to provide the mask itself as input to the model. More generally, we can represent the masked state as an absorbing state vector $\boldsymbol{a}$ which has the same shape as $\boldsymbol{x}$ but only contains a pre-specified value. The input to the network is then not the masked $\boldsymbol{m} \odot \boldsymbol{x}$, but instead the combination $\boldsymbol{m} \odot \boldsymbol{x} + (1 - \boldsymbol{m}) \odot \boldsymbol{a}$. In addition, the network $f$ may also take the time component $t$ as input as is typically done in diffusion models (Ho et al., 2020). In summary the network takes some additional inputs as $\boldsymbol{\theta} = f(\boldsymbol{i}, \boldsymbol{m}, t)$ where $\boldsymbol{i} = \boldsymbol{m} \odot \boldsymbol{x} + (1 - \boldsymbol{m}) \odot \boldsymbol{a}$ and the processing of $\boldsymbol{x}$ may be different depending on the type of data.

## 3.1 PARALLELIZED ARDMS

An important property of our parametrization is that the distribution over multiple variables is predicted at the same time. In this section, we will leverage this parameterization to allow parallel independent generation of variables. Essentially, we desire distributions over $x_{\sigma(t+k)}$ for positive $k$ while conditioning only on $\boldsymbol{x}_{\sigma(<t)}$. First we make an observation regarding a connection between predicting future variables and our likelihood terms: For $k = 1, 2, \ldots, D - t$:

$$\mathbb{E}_\sigma \big[ \log p(x_{\sigma(t+k)}|\boldsymbol{x}_{\sigma(<t)}) \big] = \mathbb{E}_\sigma \big[ \log p(x_{\sigma(t)}|\boldsymbol{x}_{\sigma(<t)}) \big] = \mathcal{L}_t, \tag{3}$$

due to the uniform expectation over permutations. In other words, it does not matter which step $t + k$ the model predicts, in expectation these all have the same associated likelihood. As a result, order agnostic generation of $k$ tokens independently, starting from the $t$-th variable will result in a log-probability contribution of $k \cdot \mathcal{L}_t$ in a single step, whereas the traditional approach would take $k$ steps at the cost of $\sum_{i=1}^{k} \mathcal{L}_{t+i}$. This knowledge is sufficient to construct a dynamic programming algorithm as described by Watson et al. (2021) to compute how many parallel steps to take at which moment, given a budget. Since dynamic programming is typically described from a minimization perspective we define the loss component $L_t = -\mathcal{L}_t$, which is measured in bits. In terms of loss, generating $k$ variables at timestep $t$ will cost $k \cdot L_t$ bits. Further, we define the transition cost matrix $\mathbf{L}_{t,t+k} = k \cdot L_t$ for positive integers $k$ and $\mathbf{L}_{t+k,t} = 0$ otherwise. So $\mathbf{L}_{t,t+k}$ exactly describes how much it costs to model the next $k$ variables in parallel starting at the $t$-th position for all relevant $t$ and $k$. Using this transition cost matrix, the dynamic programming algorithm can be utilized to find which steps should be parallelized. For instance, in the example in Figure 3 a hypothetical 20-step problem is given a budget of 5 steps. Typically, the algorithm will spend more steps on regions with large differences between $L_t$ components and fewer steps on regions where the $L_t$ components are

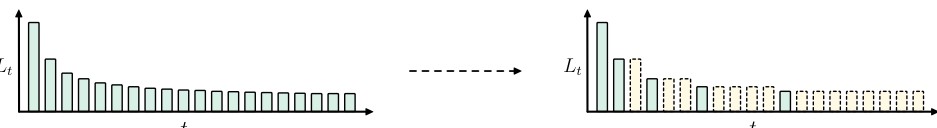

Figure 3: Loss components for Parallelized ARDMs using a budget of 5 steps for a problem of 20 steps. Left: individual loss component for every step. Right: parallelized policy extracted from the dynamic programming algorithm. Components of the same height are modelled simultaneously, so they are inferred and generated in parallel.

approximately equal. Parallelizing an ARDM may incur some cost, as for a well-calibrated model:

$$\mathcal{L}_t = \mathbb{E}_\sigma \big[ \log p(x_{\sigma(t+1)} | \boldsymbol{x}_{\sigma(<t)}) \big] \leq \mathbb{E}_\sigma \big[ \log p(x_{\sigma(t+1)} | \boldsymbol{x}_{\sigma(<t+1)}) \big] = \mathcal{L}_{t+1}, \qquad (4)$$

but can be traded off for faster generation because fewer steps are used. In other words, the *loss* components $L_t$ are monotonically *decreasing* over $t$ and parallelizing a model incurs a cost, which the algorithm aims to minimize. Recall that this is under the assumption that model is well-calibrated, which is observed in practice. See Figure 3 for an example of a parallelized schedule.

## 3.2 Depth Upscaling ARDMs

Order agnostic ARDMs learn to generate variables in random order. As a result, decisions on very detailed information (such as the least significant bit in an image) are modelled relatively early in the generative process. Instead, we can structure the process into stages, where for each stage a refinement of the variable is generated. We refer to this process as *upscaling*. For example, instead of generating an entire 256-categorical variables at once, we can first generate the most significant bit, and then the subsequent bits in order of significance. To define the process, it is helpful to first imagine the opposite process to upscaling, which is the destructive process *downscaling*. Formally, we can define maps via transition matrices $\mathbf{P}^{(i)}$ that define how a data variable downscales from its data value towards a common absorbing state. For simplicity assume single dimensional variables at this moment. Denote the absorbing state as a one-hot vector $\boldsymbol{x}^{(0)}$, where all values are zero except at a prespecified index $a$ so that $x_a^{(0)} = 1$. From a diffusion perspective, upscaling is complementary to a downscaling destruction process where each variable decays by zeroing its least significant bit.

Let $\mathbf{P}^{(1)}, \ldots, \mathbf{P}^{(S)}$ define a sequence of downscaling maps so that for any categorical one-hot data variable $\boldsymbol{x}^{(S)} \in \{0, 1\}^K$, it holds that $\mathbf{P}^{(1)} \cdot \ldots \cdot \mathbf{P}^{(S)} \cdot \boldsymbol{x}^{(S)} = \boldsymbol{x}^{(0)}$. In other words, any category $K$ decays to the common absorbing state after $S$ downscaling maps. We now define the *upscaling* generative process by learning the reverse of the downscaling map, specifically by modelling $p(\boldsymbol{x}^{(S)}|\boldsymbol{x}^{(S-1)}) \cdot \ldots \cdot p(\boldsymbol{x}^{(2)}|\boldsymbol{x}^{(1)}) p(\boldsymbol{x}^{(1)})$. The transition matrices allow easy transitions between the different stage variable $\boldsymbol{x}^{(i)}$ via the following rules:

$$\boldsymbol{x}^{(s)} = \mathbf{P}^{(s+1)} \boldsymbol{x}^{(s+1)} = \overline{\mathbf{P}}^{(s+1)} \boldsymbol{x}^{(S)}, \quad \text{where} \quad \overline{\mathbf{P}}^{(s)} = \mathbf{P}^{(s)} \cdot \mathbf{P}^{(s+1)} \cdot \ldots \cdot \mathbf{P}^{(S)}.$$

The matrices $\overline{\mathbf{P}}^{(i+1)}$ are computed as cumulative matrix multiplications, and allow a transition directly from a datapoint $\boldsymbol{x}^{(S)}$ to the corresponding downscaled variable $\boldsymbol{x}^{(i)}$. This is particularly useful during training, where the model will only be optimized for a single specific stage per datapoint. For implementations it is generally useful to define $\overline{\mathbf{P}}^{(S+1)} = \mathbf{I}$ as an identity matrix so that the above equation also holds when $s = S$. To train Upscale ARDMs, we can extend Algorithm 2: In addition to sampling a timestep $t$, a stage $i \sim \mathcal{U}(1, \ldots, S)$ to optimize is sampled. For this particular stage, the ARDM models $p(\boldsymbol{x}^{(s)}|\boldsymbol{x}^{(s-1)})$ by sampling a permutation $\sigma$ within the stage and a timestep $t$ within the stage. Every term $p(\boldsymbol{x}^{(s)}|\boldsymbol{x}^{(s-1)})$ represents a stage that is modelled with an order agnostic ARDM. This highlights an interesting property of ARDMs: Although sampling from a model may take up to $D \cdot S$ steps, the training complexity has not changed by modelling multiple stages. As a result, one can experiment with adding an arbitrary number of stages without an increase in computational complexity during training. Depth upscaling is reminiscent of the upscaling networks proposed in (Kalchbrenner et al., 2018; Menick & Kalchbrenner, 2019), with the important differences that Upscale ARDMs model the variables order-agnostic and only utilize a single neural network to parametrize all stages. For a more detailed explanation that includes the dimensionality of the variables $\{\boldsymbol{x}^{(s)}\}$ see Appendix A.

**Bit Upscaling** Depth upscaling is easiest demonstrated with an example, *bit*-upscaling. Consider the task of generating a standard image with pixel values $\{0, \ldots, 255\}$ so that an image with $D$ dimensions can be represented by $\boldsymbol{x}^{(8)} \in \{0, 1\}^{D \times 256}$ in onehot notation. Imagine a downscaling process defined by the function that removes the $i$ least significant bits: $\mathrm{lsb}_s(k) = \lfloor k/2^s \rfloor \cdot 2^s$, via this function we can define our transition matrices:

$$P_{l,k}^{(8+1-s)} = 1 \quad \text{if} \quad l = \mathrm{lsb}_s(k) \text{ and } k \in \mathrm{Im}(\mathrm{lsb}_{s-1}) \quad \text{otherwise} \quad P_{l,k}^{(8+1-s)} = 0,$$

where $\{\mathbf{P}^{(s)}\}$ are indexed starting at zero. Notice that in this case 8 stages to map every value to the absorbing state 0, because $\mathrm{lsb}_8(k) = 0$ for any $k \in \{0, \ldots, 255\}$. See Figure 4 for a visualization of such matrices for a problem with less categories.

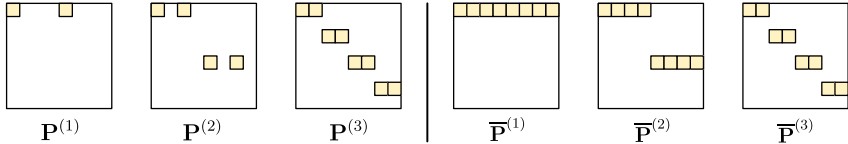

Figure 4: Bit upscaling matrices for data with eight categories and hence three stages, meaning $S = 3$. Entries that are white represent zeros, coloured entries represent ones.

Depth upscaling is not confined to bits, and indeed a more general formulation is given by the downscaling map $l = \lfloor k/b^s \rfloor \cdot b^s$, for a branching factor $b$. When $b$ is set to 2, the bit upscaling transitions are retrieved as a special case. When $b$ is set to higher values, then variables can be generated in fewer stages, $S = \lceil \log_b(K) \rceil$ to be exact. This allows for a unique trade-off between the number of steps the model takes and the complexity that each modelling step inhibits. Other hand-crafted transitions are also imaginable, not excluding transitions that augment the space to new categories, but these are not considered in this paper.

**Parametrization of the Upscaling Distributions** Although it is now defined how a datapoint $\boldsymbol{x}^{(S)}$ downscales to $\boldsymbol{x}^{(S-1)}, \ldots, \boldsymbol{x}^{(1)}$ and to its absorbing state $\boldsymbol{x}^{(0)}$, it is not immediately clear to parametrize the distributions $p(\boldsymbol{x}^{(s)}|\boldsymbol{x}^{(s-1)})$. Two methods can be used to parametrize the distribution. The first is a *direct parametrization*. In the example of the bit-upscaling model above, one models the $s$-th significant bits given the $(s-1)$-th significant bits. The direct parametrization is generally more computationally efficient, as it requires only distribution parameter outputs that are relevant for the current stage. This is especially useful when the number of classes is large (such as with audio, which has $2^{16}$ classes). However, it can be somewhat tedious to figure out exactly which classes are relevant and should be modelled.

Alternatively we can use a *data parametrization* which is similar to the parametrization in Austin et al. (2021). An important difference with their work is that the downscaling matrices $\mathbf{P}^{(s)}$ represent deterministic maps while theirs represent a stochastic process. For this parametrization, the network $f$ outputs a probability vector $\boldsymbol{\theta}$ that matches the shape of the data $\boldsymbol{x}^{(S)}$, which transformed and converted to the relevant probabilities in stage $s$ via:

$$\boldsymbol{\theta}^{(s)} = \frac{\mathbf{P}^{(s)\mathrm{T}}\boldsymbol{x}^{(s-1)} \odot \overline{\mathbf{P}}^{(s+1)}\boldsymbol{\theta}}{\boldsymbol{x}^{(s-1)\mathrm{T}}\overline{\mathbf{P}}^{(s)}\boldsymbol{\theta}} \quad \text{where} \quad p(\boldsymbol{x}^{(s)}|\boldsymbol{x}^{(s-1)}) = \mathcal{C}(\boldsymbol{x}^{(s)}|\boldsymbol{\theta}^{(s)}).$$

The advantage of this parametrization is that one only has to define the transition matrices $\{\mathbf{P}^{(s)}\}$. As a result, the appropriate probabilities can be automatically computed which is ideal for experimentation with new downscaling processes. The disadvantage may be that modelling full probability vectors for problems with high number of classes may be expensive and not even fit in memory. Empirically in our experiments on image data we find that there is no meaningful performance difference between the two parametrizations.

## 4 RELATED WORK

**Autoregressive Models** Autoregressive Models (ARMs) factorize a joint distribution into a product of conditional distributions (Bengio & Bengio, 2000; Larochelle & Murray, 2011). Advances in deep learning have allowed tremendous progress on various modalities, such as images (van den Oord et al., 2016b; Child et al., 2019, i.a.), audio (van den Oord et al., 2016a; Kalchbrenner et al., 2018, i.a.), and text (Bengio et al., 2003; Graves, 2013; Melis et al., 2018; Merity et al., 2018; Brown et al., 2020, i.a.), where for the latter they are referred to as language models.

Although evaluating the likelihood of a datapoint is generally efficient with ARMs, sampling requires an iterative process with as many network calls as the dimensionality of the data. Parallelized ARM approaches often rely either on cutting many dependencies in the conditioning (Reed et al., 2017) which tend to suffer in log-likelihood. Alternatively, ARMs can be solved using fixed-point iteration algorithms in fewer steps without sacrificing log-likelihood (Wiggers & Hoogeboom, 2020; Song et al., 2021), but these methods typically still require a large number of steps to converge.

Order agnostic sequence modelling was introduced in (Uria et al., 2014) and utilizes the same objective as AO-ARDMs to optimize the model, operating by masking and predicting variables. Different from their method, ARDMs have more choices in absorbing states, parallelization support and

Table 1: Order Agnostic model performance (in bpc) on the text8 dataset. The OA-Transformer learns arbitrary orders by permuting inputs and outputs as described in XLNet. A Transformer learning only a single order achieves 1.35 bpc.

| Model | Steps | NLL |
|---|---|---|
| OA-Transformer | 250 | 1.64 |
| D3PM-uniform | 1000 | 1.61 $_{\pm 0.020}$ |
| D3PM-absorbing | 1000 | 1.45 $_{\pm 0.020}$ |
| D3PM-absorbing | **256** | 1.47 |
| OA-ARDM (ours) | **250** | **1.43** $_{\pm 0.001}$ |
| D3PM-absorbing | **20** | 1.56 $_{\pm 0.040}$ |
| Parallelized OA-ARDM (ours) | **20** | **1.51** $_{\pm 0.007}$ |

Table 2: Order Agnostic modelling performance (in bpd) on the CIFAR-10 dataset. The upscaling model generates groups of four most significant categories, equivalent to 2 bits at a time.

| Model | Steps | NLL |
|---|---|---|
| ARDM-OA | 3072 | 2.69 $_{\pm 0.005}$ |
| Parallel ARDM-OA | 50 | 2.74 |
| ARDM-Upscale 4 | $4 \times 3072$ | **2.64** $_{\pm 0.002}$ |
| Parallel ARDM-Upscale 4 | $4 \times 50$ | 2.68 |
| D3PM Absorbing | 1000 | 4.40 |
| D3PM Gaussian | 1000 | 3.44 $_{\pm 0.007}$ |

depth upscaling techniques, in addition to modern advances to fit larger scale data. An alternative approach for order agnostic modelling is via causally masked permutation equivariant models such as Transformers (Yang et al., 2019; Alcorn & Nguyen, 2021), but these have had limited success in likelihood-based tasks. In (Ghazvininejad et al., 2019) a mask predict method is proposed, although it does not contain a likelihood analysis. In other work, mixtures of ARMs over certain orders are trained by overriding convolutional routines for masking (Jain et al., 2020). In a different context in (Liu et al., 2018) graph edges connected to a node are modelled without order. However, the model is not entirely order agnostic because it models edges centered around focus nodes.

**Diffusion Models**    Diffusion models learn to denoise a Gaussian base distribution into the distribution of the data via a chain of latent variables (Song & Ermon, 2019; Sohl-Dickstein et al., 2015; Ho et al., 2020). Diffusion and score-matching methods have shown large improvements in image (Dhariwal & Nichol, 2021) and audio sample quality (Chen et al., 2020; Kong et al., 2021), as well as likelihood improvements with variational interpretations of diffusion models (Kingma et al., 2021; Huang et al., 2021). Although faster sampling schedules for continuous diffusion models have been explored (Jolicoeur-Martineau et al., 2021; Kong & Ping, 2021), little is known about shorter generative processes for discrete diffusion.

*Discrete* diffusion models operate directly on discrete spaces. In Sohl-Dickstein et al. (2015) diffusion for binary data was proposed which was extended for categorical data in Hoogeboom et al. (2021). Whereas these approaches uniformly resample categories, in Austin et al. (2021) a wide variety of transition distributions was proposed. This work finds that absorbing diffusion produces the best performing models in log-likelihood for text data, but these models still demand a large number of steps. OA-ARDMs are equivalent to the infinite time limit of absorbing diffusion, which makes them maximally expressive. Simultaneously, ARDMs upper bound the number of steps to the dimensionality of the data. More details on the connections between these model types are in Appendix C. Other discrete diffusion processes have been explored in (Johnson et al., 2021).

## 5    RESULTS

**Order Agnostic Modelling**    To better understand how ARDMs compare to other order agnostic generative models, we study their performance on a character modelling task using the text8 dataset (Mahoney, 2011). ARDMs are compared to D3PMs that model the inverse absorbing diffusion process (Austin et al., 2021), and causally masked Transformers that are directly optimized on randomly permuted sequences as done in XLNet (Yang et al., 2019). The different methods all use the same underlying neural network architecture which is the Transformer used in (Austin et al., 2021), which has 12 layers, 786 hidden dimensions and 12 heads. For the OA-Transformer baseline the architecture is causally masked, and inputs are permuted to model the sequence in a specific order. In addition to the standard positional embeddings for the input, the embeddings for the output are also concatenated to the token embedding. This can be seen as an implicit method to condition on the permutation that is currently generated. The specific hyperparameters of the optimization procedure are specified in Appendix D and are the same as reported in (Austin et al., 2021), with the exception of a different learning rate schedule and further train steps.

Performance of these methods is presented in Table 1. Firstly, the OA-Transformer baseline does not perform very well compared to the other models. This result matches the behaviour that was found

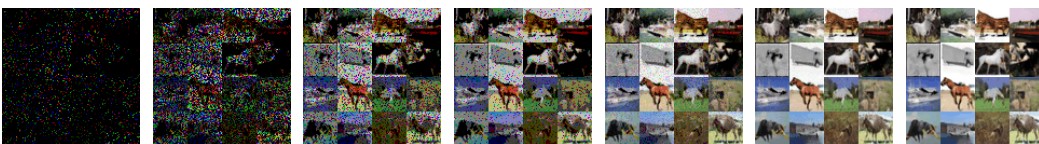

Figure 5: Visualization of $x$ through the generative process for an ARDM Upscale 4 model.

by Yang et al. (2019), who observed underfitting behaviour and limited the task complexity by only predicting a subset of the permuted tokens. Further, as expected the performance of our OA-ARDM with 1.43 bpc is very close to the performance of D3PM-absorbing at 1000 steps with 1.45 bpc. This is expected, since OA-ARDMs are equivalent to the continuous time limit of D3PM-absorbing models. For sequences containing only 250 dimensions, the D3PM schedule with 1000 steps starts to approximate the jump process where generally only a single variable is absorbed at a time. The important takeaway from this comparison is that OA-ARDMs perform similar to large-steps D3PM absorbing models *while only requiring a quarter of the steps*. When the D3PM model is forced to take 256 steps which is comparable to our OA-ARDM model, then its performance degrades further towards 1.47 bpd. In addition, a Parallelized ARDM with only 20 steps has a performance of 1.51 bpd over a similar D3PM which has 1.56 bpd. This pattern translates to CIFAR-10 (Krizhevsky et al., 2009) where ARDMs also outperform D3PMs and degrade more gracefully under fewer steps. This comparison to D3PM is however less direct, as the underlying architectures differ.

**Lossless Compression**    To validate that ARDMs can form a viable basis for practical neural network-based compressors, we study their performance when compressing CIFAR-10 images and comparing them to existing methods. Since ARDMs provide probabilities for a sequence of symbols, they can be directly used together with an off-the-shelf entropy coder for lossless compression. In this experiment we use the range-based entropy coder rANS (Duda, 2009). To use ARDMs the order of the coding process needs to be fixed for all images. To avoid an unlucky sample, before coding we evaluate the log-likelihood of a few random permutations on the train set and pick the best performing one. Empirically, there is very little difference in performance ($< 0.02$ bpd) between different permutations.

Several deep learning based lossless compression methods in literature rely on bits-back coding (Townsend et al., 2019), such as LBB (Ho et al., 2019), HiLLoC (Townsend et al., 2020) and VDM (Kingma et al., 2021). Although bits-back coding methods can perform well on large datasets, they have a large overhead when used as per-image compressors. This is caused by the large number of initial bits that are required. Further, the dataset is often interlinked, meaning that if an image in the middle of the dataset needs to be accessed, it requires all images earlier in the bitstream to also be decompressed. Therefore per-image compression is important for practical applications, because it is desirable to be able to send a specific image without sending an entire dataset. On the other hand, direct compressors such as L3C (Mentzer et al., 2019), IDF (Hoogeboom et al., 2019) and IDF++ (van den Berg et al., 2021) do not incur an intial message overhead and their dataset performance translates directly to per-image compression. A more conventional codec is FLIF (Sneyers & Wuille, 2016), which is a recent lossless compression codec with machine learning components that outperforms traditional codecs such as PNG.

Performance of ARDMs and related methods in literature is presented in Table 3. ARDMs significantly outperform all methods on compression per image, requiring only 2.71 bpd versus 3.26 for the next best performing model, IDF++. In addition, even compared to a setting where an entire dataset needs to be compressed, ARDMs perform competitively to VDM, which attain 2.72 bpd. Moreover, ARDMs degrade more gracefully when fewer steps are used to encode the data.

Note that the lossless compressor based on VDM was trained on non-augmented data, whereas the best-performing likelihood model of Kingma et al. (2021) was trained with data augmentation. As a result, it is likely that their dataset compression results could be somewhat improved when trained on augmented CIFAR-10. Also, it is not a coincidence that HiLLoC and FLIF have the exact same compression per image performance. HiLLoC compresses the first few images using the FLIF format to fill the initial bitstream, and compresses the remaining images in the dataset with bits-back coding (Townsend et al., 2020). As a result, on a per-image compression benchmark the method is equivalent to FLIF.

**Effects of Depth-Upscaling**    A natural question that might arise is how standard order-agnostic modelling performs compared to order agnostic bit-upscaling, and how bit-upscaling compares to

Table 3: CIFAR-10 lossless compression performance (in bpd).

| Model | Steps | Compression per image | Dataset compression |
|---|---|---|---|
| VDM (Kingma et al., 2021) | 1000 | $\geq 8$ | 2.72 |
| VDM (Kingma et al., 2021) | 500 | $\geq 8$ | 2.72 |
| OA-ARDM (ours) | 500 | 2.73 | 2.73 |
| ARDM-Upscale 4 (ours) | 500 | **2.71** | **2.71** |
| VDM (Kingma et al., 2021) | 100 | $\geq 8$ | 2.91 |
| OA-ARDM (ours) | 100 | 2.75 | 2.75 |
| ARDM-Upscale 4 (ours) | 100 | 2.76 | 2.76 |
| LBB (Ho et al., 2019) | | $\geq 8$ | 3.12 |
| IDF (Hoogeboom et al., 2019) | | 3.34 | 3.34 |
| IDF++ (van den Berg et al., 2021) | | 3.26 | 3.26 |
| HiLLoC (Townsend et al., 2020) | | 4.19 | 3.56 |
| FLIF (Sneyers & Wuille, 2016) | | 4.19 | 4.19 |

Table 4: Audio (SC09) depth upscaling test set performance (in bpd). A WaveNet baseline learning only a single order achieves 7.77 bpd.

| Model | Steps | Performance |
|---|---|---|
| OA-ARDM | $D = 16000$ | 7.93 |
| ARDM Upscale 256 | $2 \times D$ | 6.36 |
| ARDM Upscale 16 | $4 \times D$ | 6.30 |
| ARDM Upscale 4 | $8 \times D$ | **6.29** |
| ARDM Upscale 2 | $16 \times D$ | **6.29** |

Table 5: Image (CIFAR-10) depth upscaling performance (in bpd).

| Model | Steps | Performance |
|---|---|---|
| OA-ARDM | $D = 3072$ | 2.69 |
| ARDM Upscale 16 | $2 \times D$ | 2.67 |
| ARDM Upscale 4 | $4 \times D$ | **2.64** |
| ARDM Upscale 2 | $8 \times D$ | 2.67 |

the upscaling with larger values. Due to the constant training complexity of ARDMs, one can easily train models that have generative processes of arbitrary length. To test this, we train ARDMs on image data from CIFAR-10 and audio data from SC09 (Warden, 2018). For the audio data, the total number of categories is $2^{16}$, which is typically too large in terms of memory to model as a single softmax distribution. For that reason, the single stage OA-ARDM is trained using a discretized logistic distribution because it is computationally cheaper for a high number of categories. For the same reason, the Upscale ARDMs for audio can only be trained using the direct parametrization, whereas for images they are trained with the data parametrization.

For images, the best performing model has an upscaling factor of 4 with 2.64 bpd (see Table 5) and for audio the best performing model upscales by a factor of 2 or 4 with 6.29 bpd (see Table 4). The hypothesis is that as the upscale factor becomes smaller, the generative process generally becomes more structured and easier to model. However, although for audio this pattern is consistently observed, for images an upscale factor of 4 has better performance than an upscale factor of 2. It is possible that for certain data, at some point smaller upscale factors give diminishing returns for performance. We hypothesize that by prolonging the generative process, the model may get less gradient signal per optimization step, leading to the decreased performance of smaller upscale factors in some situations.

## 6 LIMITATIONS AND CONCLUSION

Notwithstanding the good results in this paper, there are some limitations to ARDMs. 1) Even though ARDMs outperform all other order-agnostic approaches on text, there is still a gap to the performance of single-order autoregressive models. In preliminary experiments, upscale variants for language did not perform better than the order-agnostic versions. 2) In the current description, ARDMs model discrete variables. In principle one could also define absorbing processes for continuous distributions. 3) Finally, in this work we have focused on optimizing for log-likelihood, because it directly corresponds to coding length in lossless compression. However when optimizing for other objectives such as sample quality, different architectural choices may give better results.

In conclusion, we introduced ARDMs, a new class of models at the intersection of autoregressive models and discrete diffusion models. ARDMs perform competitively with existing generative models, and outperform competing approaches on per-image lossless compression.

## REPRODUCIBILITY AND ETHICS STATEMENT

To ensure the work is as reproducible as possible, in this paper we have described in detail both the training algorithms and the sampling algorithms. The main ideas are presented in Section 3, and further clarifications that may be important for re-implementation are given in Appendix A. The hyperparameter settings to run experiments are presented in Section 5 and further clarified in Appendix D. In addition, we plan to release the code that can be used to reproduce the experimental results in this paper.

In terms of ethics, we do not see immediate concerns for the models we introduce. However, deep generative models have a wide variety of applications such as representation learning, image inpainting, special effects in video, outlier detection and drug design. On the other hand, the generation of images, text and video may have negative downstream applications such as making false media seem realistic. Further, to the best of our knowledge no datasets were used that have known ethical issues.

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

## A    FURTHER DETAILS OF AUTOREGRESSIVE DIFFUSION

Next to given descriptions, the implementation has been open-sourced at `https://github.com/google-research/google-research/tree/master/autoregressive_diffusion`.

### A.1    DEPTH UPSCALING

This section explains further details that are important to optimize and sample from Depth Upscaling ARDMs, which are summarized in Algorithm 3 and 4. Recall that for depth-upscaling models, the variables are modelled in stages $\boldsymbol{x}^{(1)}, \ldots, \boldsymbol{x}^{(S)}$ and the model learns $p(\boldsymbol{x}^{(S)}|\boldsymbol{x}^{(S-1)}), \ldots, p(\boldsymbol{x}^{(1)}|\boldsymbol{x}^{(0)})$. Here $\boldsymbol{x}^{(0)}$ is a constant absorbing state and $\boldsymbol{x}^{(S)}$ represents the data. The transition matrices $\{\mathbf{P}^{(s)}\}$ describe the destructive maps which end up in the absorbing state. They form the destructive counterpart of the generative process.

Instead of optimizing for all stages simultaneously, we sample a stage uniformly $s \sim \mathcal{U}(1, \ldots, S)$ and optimize for that stage. Here the cumulative matrix products $\overline{\mathbf{P}}^{(s)}$ allow us to directly transition to a specific stage, since $\boldsymbol{x}^{(s)} = \overline{\mathbf{P}}^{(s+1)}\boldsymbol{x}^{(S)}$. To be precise, for a single dimension $i$ the variable $\boldsymbol{x}_i^{(s)}$ is represented as a onehot vector and then transformed using the matrix multiplication $\boldsymbol{x}_i^{(s)} = \overline{\mathbf{P}}^{(s+1)}\boldsymbol{x}_i^{(S)}$. For multiple dimensions this matrix multiplication is applied individually, meaning that $\overline{\mathbf{P}}^{(s+1)}\boldsymbol{x}^{(S)} = \big(\overline{\mathbf{P}}^{(s+1)}\boldsymbol{x}_1^{(S)}, \overline{\mathbf{P}}^{(s+1)}\boldsymbol{x}_2^{(S)}, \ldots, \overline{\mathbf{P}}^{(s+1)}\boldsymbol{x}_D^{(S)}\big) = (\boldsymbol{x}_1^{(s)}, \ldots, \boldsymbol{x}_D^{(s)}) = \boldsymbol{x}^{(s)}$.

For a optimization step, a stage $s \sim \mathcal{U}(1, \ldots, S)$ and a step $t \sim \mathcal{U}(1, \ldots, D)$ are sampled, in addition to a permutation $\sigma \sim \mathcal{U}(S_D)$. Then using the cumulative matrices, from a datapoint $\boldsymbol{x} = \boldsymbol{x}^{(S)}$ the variables $\boldsymbol{x}^{(s)}$ and $\boldsymbol{x}^{(s-1)}$ are computed. As before, the mask $\boldsymbol{m} = \sigma < t$ gives the locations of the variables that are conditioned on. For those locations the values in $\boldsymbol{x}^{(s)}$ may already be accessed. For the opposite locations $1 - \boldsymbol{m}$, instead the values from $\boldsymbol{x}^{(s-1)}$ are accessed. This leads to the expression for the input $\boldsymbol{i} = \boldsymbol{m} \odot \boldsymbol{x}^{(s)} + (1-\boldsymbol{m}) \odot \boldsymbol{x}^{(s-1)}$. The target of the network will be to predict a distribution for $\boldsymbol{x}^{(s)}$ at the locations at $1 - \boldsymbol{m}$. The network will take in the computed input $\boldsymbol{i}$ together with variables to clarify in which stage of the generative process the model is, $\boldsymbol{m}$, $s$ and $t$. In case of the data parametrization, the probabilities $\boldsymbol{\theta}$ are appropriately normalized and reweighted to $\boldsymbol{\theta}^{(s)}$ using transitions $\{\mathbf{P}^{(s)}\}$. Then, the log probabilities $\log \mathcal{C}(\boldsymbol{x}^{(s)}|\boldsymbol{\theta}^{(s)})$ are computed elementwise over dimensions and subsequently masked with $1 - \boldsymbol{m}$. These quantities are then summed and reweighted to get a stochastic estimate for the ELBO.

For the sampling, the model traverses through each stage, and for each stage through every dimension in a different order. In each step the network together with the transition matrices produces a probability vector $\boldsymbol{\theta}^{(s)}$ from which elementwise samples are taken $\boldsymbol{x}' \sim \mathcal{C}(\boldsymbol{x}^{(s)}|\boldsymbol{\theta}^{(s)})$, but only the values at locations $\boldsymbol{n} \leftarrow (\sigma = t)$ are filled in, corresponding to the current generation step. By traversing through all steps and stages, the variable $\boldsymbol{x}^{(S)}$ is generated.

---

**Algorithm 3** Sampling from Upscale-ARDMs

> **Input:** Network $f$
> **Output:** Sample $\boldsymbol{x}$
> Initialize $\boldsymbol{x} = \boldsymbol{x}^{(0)}$
> **for** $s$ in $\{1, \ldots, S\}$ **do**
>     Sample $\sigma \sim \mathcal{U}(S_D)$
>     **for** $t$ in $\{1, \ldots, D\}$ **do**
>         $\boldsymbol{m} \leftarrow \sigma < t$
>         $\boldsymbol{n} \leftarrow (\sigma = t)$
>         $\boldsymbol{\theta} \leftarrow f(\boldsymbol{x}, \boldsymbol{m}, s, t)$
>         $\boldsymbol{\theta}^{(s)} \propto \mathbf{P}^{(s)^{\mathrm{T}}}\boldsymbol{x} \odot \overline{\mathbf{P}}^{(s+1)}\boldsymbol{\theta}$
>         $\boldsymbol{x}' \sim \mathcal{C}(\boldsymbol{x}^{(s)}|\boldsymbol{\theta}^{(s)})$
>         $\boldsymbol{x} \leftarrow (1 - \boldsymbol{n}) \odot \boldsymbol{x} + \boldsymbol{n} \odot \boldsymbol{x}'$

**Algorithm 4** Optimizing Upscale-ARDMs

> **Input:** Datapoint $\boldsymbol{x}$, Network $f$
> **Output:** ELBO $\mathcal{L}$
> Sample $s \sim \mathcal{U}(1, \ldots, S)$
> Sample $t \sim \mathcal{U}(1, \ldots, D)$
> Sample $\sigma \sim \mathcal{U}(S_D)$
> $\boldsymbol{x}^{(s)} \leftarrow \overline{\mathbf{P}}^{(s+1)}\boldsymbol{x}$ and $\boldsymbol{x}^{(s-1)} \leftarrow \overline{\mathbf{P}}^{(s)}\boldsymbol{x}$
> Compute $\boldsymbol{m} \leftarrow \sigma < t$
> $\boldsymbol{i} \leftarrow \boldsymbol{m} \odot \boldsymbol{x}^{(s)} + (1 - \boldsymbol{m}) \odot \boldsymbol{x}^{(s-1)}$
> $\boldsymbol{\theta} \leftarrow f(\boldsymbol{i}, \boldsymbol{m}, s, t)$
> $\boldsymbol{\theta}^{(s)} \propto \mathbf{P}^{(s)^{\mathrm{T}}}\boldsymbol{x}^{(s-1)} \odot \overline{\mathbf{P}}^{(s+1)}\boldsymbol{\theta}$
> $\boldsymbol{l}_t \leftarrow (1 - \boldsymbol{m}) \odot \log \mathcal{C}(\boldsymbol{x}^{(s)}|\boldsymbol{\theta}^{(s)})$
> $\mathcal{L} \leftarrow \frac{D}{D-t+1} \mathrm{sum}(\boldsymbol{l}_t)$

---

## A.2 Details on Parallelized ARDMs

This section discusses further details on Parallelized ARDMs, and provides a JAX version of the dynamic programming algorithm from (Watson et al., 2021) that was written in NumPy. Since the algorithm scales with $\mathcal{O}(D^3)$ this implementation is important to scale to larger dimensional problems. To clarify, the upscale ARDMs can be seen as a $S$ sequential OA-ARDMs that model $p(\boldsymbol{x}^{(s)}|\boldsymbol{x}^{(s-1)})$, and when a parallel schedule is computed, it is computed for each stage separately. It is also possible to run the dynamic programming algorithm for all $S \cdot D$ steps simultaneously, which could even choose to distribute steps unevenly over stages, but that is not done in this paper.

Recall that to run the algorithm a matrix $\mathbf{L}$ is needed which gives the cost of travelling from one generation step to another. It is constructed so that $L_{t,t+k} = k \cdot L_t$ for positive $k$ and 0 otherwise, which represents the cost of generating $k$ variables in parallel where $L_t$ is the loss component. In practice this is implemented via a cumulative sum of a triangular mask. This part is relatively computationally cheap.

```
import jax
from jax import numpy as jnp
import numpy as np

def get_nelbo_matrix(loss_components: np.ndarray):
  num_timesteps = len(loss_components)

  # Creates multiplicative mask. E.g. if num_timesteps = 3 then:
  #         [1 2 3]
  # triu = [0 1 2].
  #         [0 0 1]
  triu = np.triu(np.ones((num_timesteps, num_timesteps)))
  triu = np.cumsum(triu[::-1], axis=0)[::-1]

  # Compute nelbos[s, t] which contains -logp(x_s | x_t)
  nelbos_ = loss_components[:, None] * triu
  # Pad last row / first column.
  nelbos = np.zeros((num_timesteps + 1, num_timesteps + 1))
  nelbos[:-1, 1:] = nelbos_

  return nelbos
```

The most expensive part of the algorithm is the loop which has computational complexity $\mathcal{O}(D^3)$. This is the most important extension of the NumPy version and reduces runtime from 5 minutes to about 2 seconds for $D = 3072$, which would be very impractical to run for our audio experiments where $D = 16000$, which now take less than half a minute to run. Through JAX this loop is XLA-compiled with the scan operation, limiting overhead when running the algorithm.

```
@jax.jit
def inner_cost_and_dimension_loop(
    nelbos: jnp.ndarray, first_cost: jnp.ndarray):
  """Inner jax-loop that computes the cost and dimension matrices."""
  num_timesteps = first_cost.shape[0] - 1

  def compute_next_cost(prev_cost: jnp.ndarray, _: jnp.ndarray):
    bpds = prev_cost[:, None] + nelbos
    new_dimension = jnp.argmin(bpds, axis=0)
    new_cost = jnp.min(bpds, axis=0)
    return new_cost, (new_cost, new_dimension)

  _, (costs, dimensions) = jax.lax.scan(
      compute_next_cost, init=first_cost,
      xs=jnp.arange(1, num_timesteps+1))

  return costs, dimensions
```

The inner algorithm logic is then called via the function below. It first builds the loss transition matrix **L** which is referred to as `nelbos` and then calls the inner loop. As an output it gives the cost and dimension matrices that can be used to *1)* find an optimal path and *2)* describe how expensive such paths are. As can be seen in Figure 6, the running average of the loss components $\{L_t\}$ might be somewhat noisy, which can negatively influence the algorithm. As a straightforward method to reduce variance of the values $\{L_t\}$, they are *sorted* before they are given to the algorithm. This is uniquely possible for ARDMs, as we expect $L_t$ to be monotonically decreasing over $t$ (see also Equation 4). For Upscale ARDMs that have multiple stages, the loss components are seperately sorted per stage.

```python
def get_cost_and_dimension_matrices(loss_components: np.ndarray):
  """Compute cost and assignment matrices, in JAX."""
  num_timesteps = len(loss_components)

  # First row of the costs matrix.
  first_cost = np.full((num_timesteps + 1,), np.inf)
  first_cost[0] = 0
  first_cost = jnp.array(first_cost)

  # First row of the dimensions matrix. The first row just contains -1
  # and is never used, but this way it aligns with the cost matrix.
  first_dimension = jnp.full((num_timesteps + 1), -1, dtype=np.int32)

  # nelbos[s, t] is going to contain the value logp(x_s | x_t)
  nelbos = jnp.array(get_nelbo_matrix(loss_components))
  costs, dimensions = inner_cost_and_dimension_loop(nelbos, first_cost)

  # Concatenate first rows to the matrices.
  costs = jnp.concatenate([first_cost[None, :], costs], axis=0)
  dimensions = jnp.concatenate([first_dimension[None, :], dimensions],
                                axis=0)

  costs = np.array(costs)
  dimensions = np.array(dimensions)

  return costs, dimensions
```

The final part of this algorithm is used to retrieve the path that needs to be taken to attain a certain cost. This algorithm takes as input a budget and the cost & dimension matrices, and returns the corresponding path to traverse.

```python
def get_optimal_path_with_budget(budget: int, costs: np.ndarray,
                                   dimensions: np.ndarray):
  num_timesteps = len(costs) - 1
  t = num_timesteps
  path = np.zeros(budget, dtype=np.int32)
  cost = costs[budget, num_timesteps]
  for k in reversed(range(1, budget+1)):
    t = dimensions[k, t]
    path[k-1] = t
  return path, cost
```

# B ADDITIONAL RESULTS

## B.1 RELATION TO OTHER LIKELIHOOD-BASED GENERATIVE MODELS

In this section we show how ARDMs perform compared to existing likelihood based generative models in literature. These results are presented in Table 6. The best performing model is the Variational Diffusion Model (VDM) (Kingma et al., 2021). ARDMs perform competitively with a best score of $2.64$ bpd, and are the best performing model among discrete diffusion approaches.

Table 6: CIFAR-10 generative modelling.

| Model | Type | NLL |
|---|---|---|
| ARDM-AO (ours) | Discrete Diffusion $\cup$ ARM | 2.69 |
| ARDM-Upscale 4 (ours) | Discrete Diffusion $\cup$ ARM | 2.64 |
| D3PM Gaussian (Austin et al., 2021) | Discrete Diffusion | 3.44 |
| DDPM (Ho et al., 2020) | Diffusion | 3.69 |
| Improved DDPM (Nichol & Dhariwal, 2021) | Diffusion | 2.94 |
| VDM (Kingma et al., 2021) | Diffusion | **2.49** |
| PixelCNN++ (Salimans et al., 2017) | ARM | 2.92 |
| SPN (Menick & Kalchbrenner, 2019) | ARM | 2.90 |
| Sparse Transformer (Jun et al., 2020) | ARM | 2.52 |
| NVAE (Vahdat & Kautz, 2020) | VAE | 2.91 |
| Very Deep VAE (Child, 2021) | VAE | 2.87 |
| CR-VAE (Sinha & Dieng, 2021) | VAE | 2.52 |

## B.2 ADDITIONAL AUDIO EXPERIMENTS

In Table 7 we present additional experimental results from our best Upscale ARDM model for the SC09 dataset (branching factor $4$), in which we consider smaller computational budgets. Recall that dimensionality $D = 16000$ for SC09 data.

## B.3 LOSS COMPONENTS OVER TIME

Since the training algorithm estimates the NLL by sampling a step $t$ for each input in the batch, we can collect and keep track of the loss components $\{L_t\}$ and plot them as a function of $t$ (see Figure 6). These are collected by updating an exponential moving average during training, and are used in the dynamic programming routine. As expected by Equation 4, the components $L_t$ are monotonically decreasing over the step $t$ within a stage. The height is re-normalized so that the average height represents the total bits per dimension. As a result, in the upscale model the value divided by number of stages $S$ represents the actual uncertainty of generating that token.

The loss plot of the upscale model allows for interesting observations: For instance, After an initially high uncertainty ($\approx 8/S = 2$ bits) the most significant bits become increasingly easier to model very fast ($< 2/S = 0.5$ bits). In contrast, for each stage that follows, the average height of that stages increases. This indicates that the model is more uncertain for the less significant bits. This can be a combination of two things: The model may have more difficulty in modelling these less significant

Table 7: Audio (SC09) depth upscaling test set performance (in bpd) for various computational budgets.

| Model | Steps | Performance |
|---|---|---|
| ARDM Upscale 4 | $8 \times 16000$ | **6.29** |
| ARDM Upscale 4 | $8 \times 1000$ | 6.30 |
| | $8 \times 500$ | 6.30 |
| | $8 \times 100$ | 6.32 |
| | $8 \times 50$ | 6.32 |

bits (i.e. high KL between data and model distribution), and the data distribution may be more uncertain in those regions (i.e. high entropy of the data distribution).

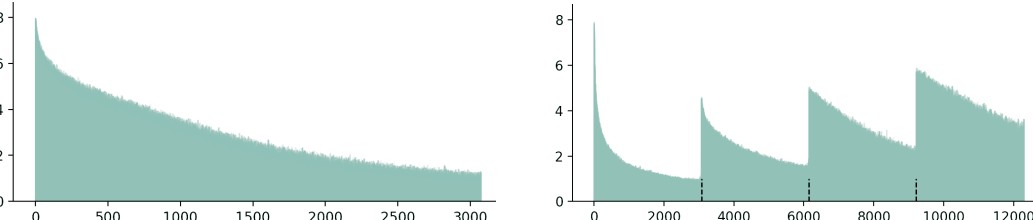

Figure 6: Loss components over model step on CIFAR-10. The height is normalized so that the average represents the total bits per dimension. Left: loss terms for the OA-ARDM. Right: loss terms for the ARDM-Upscale 4, which comprises four stages.

## B.4  SAMPLES FROM ARDMS

**Language**  Sampling from ARDMs can be visualized at different steps $t$ to highlight the generative process. Recall that for models trained on language, the absorbing state augments the space, meaning that an additional index that is added for the absorbing state. We visualize this token by the underscore character '_'. The process at four selected steps in the process are presented in Figure 7, where the last sentence represents the resulting sample.

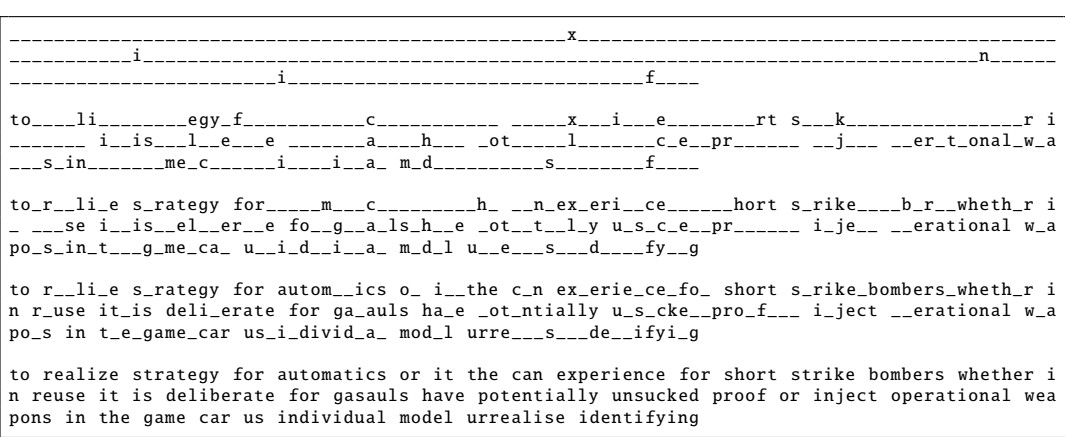

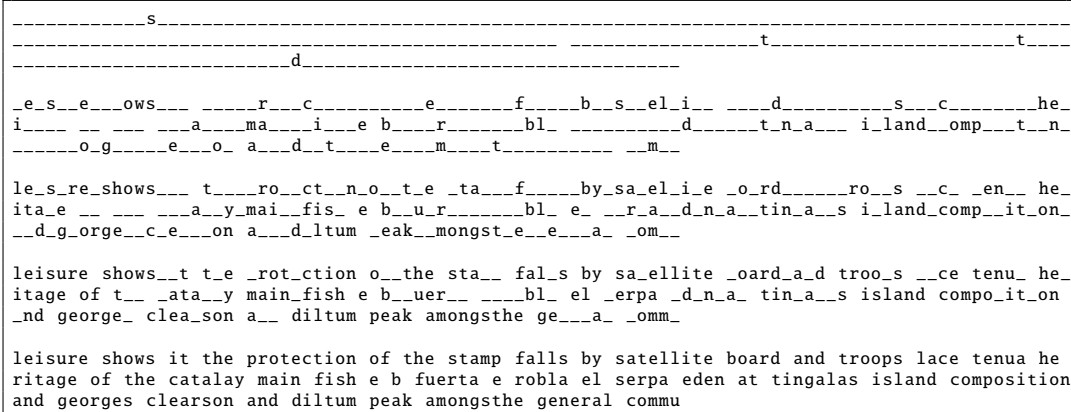

Figure 7: Two generative processes of an OA-ARDM trained on text8. The resulting sample from the model is at the bottom of each frame.

**Images**   The generative processes for images are visualized in Figure 8. In constrast with the language model, here the absorbing state takes on a specific value in the domain of the image itself. In the case of OA-ARDMs, the absorbing state is $128$ so that it is $0$ when normalized in the network architecture. In constrast, the absorbing state of the Upscale ARDM is $0$ because it is defined by zeroing least significant bits until everything is zero. The right-most grid represents the resulting samples from the model. The generative processes are very different: whereas the upscale ARDM first generates a coarses version of the images with fewer bits, the order agnostic ARDM generates each value at once.

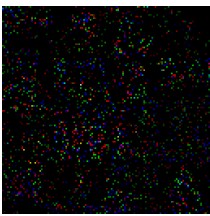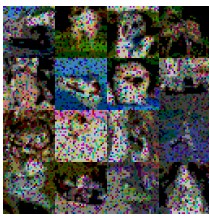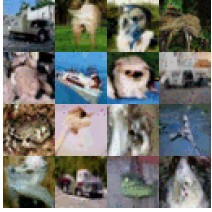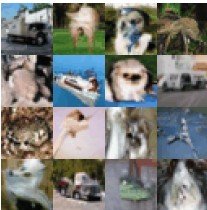

(a) Generative process of an Upscale 4 ARDM. This model was trained without data augmentation and with dropout, which explains that all images are generated upright. The performance of this model is approximately 2.71 bpd whereas the same model trained with data augmentation has 2.64 bpd.

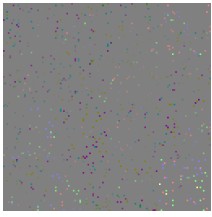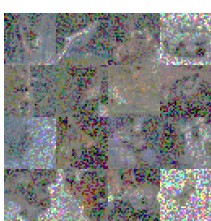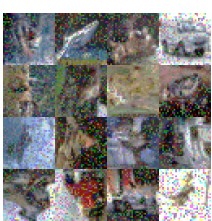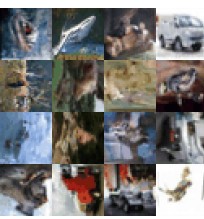

(b) Generative process of an OA-ARDM, starting at the absorbing state $\boldsymbol{a}$. This model was trained with data augmentation, which is known to somewhat degrade sample quality and naturally sometimes samples rotated or reflected images.

Figure 8: Visualization of the generative process for $\boldsymbol{x}$, ending with the resulting samples at the right-most grid.

## C   EQUIVALENCE OF AO-ARDMS AND ABSORBING DIFFUSION IN CONTINUOUS TIME

In this section we examine the connection between absorbing diffusion and AO-ARDMs more closely. We start with a description of the independent process of absorbing diffusion as used in (Austin et al., 2021) and highlight potential complications of this process. Then, we will show that AO-ARDMs are equivalent to a continuous-time version of absorbing diffusion models.

**The Independent Absorbing Process from Austin et al.**
In absorbing diffusion as described by (Austin et al., 2021), each dimension can independently get absorbed with some small probability for each time step. Specifically, letting a vector $\boldsymbol{x}(t)$ represent a Markov process as a collection of random variables index by integers $t$, where $\boldsymbol{x}(0)$ is the data distribution. Each dimension $x_i(t)$ as an equal and independent chance of getting absorbed according to rate $\gamma(t)$ at index $t$ to the absorbing state $a_i$. Define the cumulative chance of retaining the state as $\alpha(t) = \prod_{\tau=1}^{t}(1 - \gamma(\tau))$. This allows the direct expression for the distribution over $x_i(t)$ as categorical on data and the absorbing state $\{x_i(0), a_i\}$ with probabilities $\{\alpha(t), 1 - \alpha(t)\}$. Typically, the decay rate $\gamma$ is chosen so that $\alpha(T) = 0$ for some large integer $T$. For example in (Austin et al., 2021) it is set $T = 1000$ for most experiments. We refer to the absorbing process from (Austin et al., 2021) as an *independent* absorbing process, due to its independent absorbing probabilities between dimensions.

The reverse of this absorbing process is the generative process. As described above, the chance of a dimension absorbing is independent. As a result when $T$ is small, it is inevitable that multiple dimensions decay at once. This has a direct consequence for the generative process, which is parametrized to model dimensions independently. The generative process will have to model the variables of these multiple absorbed dimensions as an independent factorized distribution, which causes a loss in modelling performance. This problem can be overcome by setting $T$ to a larger value. Indeed, when $T$ is larger the chance of multiple dimensions decaying at once decreases. During training, $T$ can be set arbitrarily high without incurring costs. However, to sample or evaluate the likelihood of a specific datapoint, the computational cost scales directly with $T$ so it is desired to keep $T$ as low as possible.

As an example, consider the experiment from (Austin et al., 2021) where text sequences of length 256 are modelled using a 1000 timestep absorbing diffusion model. When sampling from this model, at least 744 of the neural network forward passes do nothing to the latent variable and are needless compute. When $T$ is reduced, performance degrades. In addition, it can be difficult to determine beforehand how high a $T$ should be sufficient, and it depends on the data and the decay rate.

**ARDMs model the reverse of a Fixed Absorbing Process**
Our ARDMs can be viewed as learning the generative process of a slightly modified absorbing process. Instead of independent absorbing costs, exactly one dimension decays at a time step until all dimensions have been absorbed. Since only one variable decays at a time, we refer to this process as a *fixed* absorbing process. This ensures that $T = D$ exactly, where $D$ is the dimensionality of the data.

An equivalent way to describe this process is by sampling a permutation of the indices $1, \ldots, D$ and decaying in that order towards the absorbing state. The corresponding generative process is then modelling the variables exact opposite order of the permutation: an AO-ARDM. As a result the generative process with a fixed absorbing process process only requires at most $D$ steps.

**ARDMs are equivalent to Continuous Time Absorbing Models**
The absorbing diffusion model from (Austin et al., 2021) can be relaxed to continuous time. Define a continuous-time discrete-space absorbing-state Markov process as a collection of Markov random variables $\{\mathbf{x}(t)\}$ in dimension $D$, parameterized by $t \in \mathbb{R}^+$. Starting in state $\boldsymbol{x}(0)$, each of the elements $x_i(t)$ of the state vector independently decays towards an absorbing state $\boldsymbol{a}$ at rate $\gamma(t)$, such that at time $t$ the distribution on the vector elements is categorical on $\{x_i(0), a_i\}$, with probabilities $\{\alpha(t), 1 - \alpha(t)\}$, with $\alpha(t) = \exp(-\int_0^t \gamma(s)ds)$. This last equivalence is obtained via the first order logarithmic Taylor expansion $\log(1 - x) \approx -x$ which holds for the small $\gamma(s)ds$.

An equivalent way of describing this stochastic process is as a finite set of $D$ random transition times $\{\tau_i\}$ for $i \in \{1, \ldots, N\}$ describing the time where the element $x_i$ has transitioned into the absorbing

state. Specifically, we say that $\tau_i$ is the latest time for which $x_i$ is non-yet absorbed, so $x_i(t) = a_i$ for $t > \tau_i$. From this perspective, $\boldsymbol{x}$ is only changing at the transition times $\tau_i$ and remains the same at other times. Then, the reverse process to model $\{\boldsymbol{x}(t)\}$ for all $t$ is equivalent to only modelling the finite dimensional $\{x_i(\tau_i), \tau_i\}$. In other words, to model the reverse process, we only need to model the transition times $\{\tau_i\}$ and the variable right before it was absorbed.

To show an equivalence between the continuous time absorbing process and ARDMs, we will show that we can model the reverse process given by $\{x_i(\tau_i), \tau_i\}$ by sampling the transition times independently, and by using an AO-ARDM for the transitions in $\boldsymbol{x}$.

The distributions $\{\tau_i\}$ are already known, they are given by the distribution with the cumulative distribution function $1 - \alpha(t)$. That leaves the modelling of $\{x_i(\tau_i)\}$ to model the reverse process. An important question that now arises is whether the transition times $\{\tau_i\}$ provide any additional information in modelling the variables $\{x_i(\tau_i)\}$. Apart from knowing *that* the variable will be un-absorbed, the answer is no. This is because the values are distributed as the data distribution so $x_i(0)|\boldsymbol{x}(t) \sim x_i(\tau_i)|\boldsymbol{x}(t)$ and the actual continuous value of $\tau_i$ does not matter, specifically $\{x_i(0)\} \perp \{\tau_i\}|\boldsymbol{x}(t)$.

As a consequence, the model for the reverse process does not need to be conditioned on the precise values $\{\tau_i\}$ and can instead be solely conditioned on $x_i(\tau_{i+1})$ to model $x_i(\tau_i)$ for all dimensions $i$. Recall that this process is equivalent to the generative process of our AO-ARDM: Each new timestep, a new dimension of the variable is modelled. The order in which the variables are modelled depends on the decay times $\{\tau_i\}$, and since these are all identically distributed, the order is uniform over all possible permutations.

We claim that we can write the VLB as follows:

$$\log p(\boldsymbol{x}(0)) \geq \mathbb{E}_{q(\boldsymbol{x}(>0)|\boldsymbol{x}(0))} \log p(\boldsymbol{x}(>0)) + \log p(\boldsymbol{x}(0)|\boldsymbol{x}(>0)) - \log q(\boldsymbol{x}(>0)|\boldsymbol{x}(0))$$

$$= \mathbb{E}_{q(\tau_1,\ldots,\tau_D)} \sum_i \log p(\boldsymbol{x}(\tau_i)|\boldsymbol{x}(\tau_{i+1})) - \mathrm{KL}(q(\tau_1,\ldots,\tau_D)|p(\tau_1,\ldots,\tau_D))$$

$$= \mathbb{E}_{\sigma \sim \mathcal{U}(S_d)} \sum_i \log p(\mathbf{x}_{\sigma(i)}|\boldsymbol{x}_{\sigma(<i)}),$$

where $\boldsymbol{x}(> 0)$ denotes all values of $\boldsymbol{x}(t)$ for $t > 0$. The first equivalence follows from the above described equivalent representation of the continuous process. And indeed when the transition times and the values of $\boldsymbol{x}$ at the transition times are given, the remaining variables of the continuous process can be reconstructed so it can be ensured that:

$$\mathrm{KL}(q(\boldsymbol{x}|\boldsymbol{x}(\tau_1),\ldots,\boldsymbol{x}(\tau_D),\tau_1,\ldots,\tau_D)|p(\boldsymbol{x}|\boldsymbol{x}(\tau_1),\ldots,\boldsymbol{x}(\tau_D),\tau_1,\ldots,\tau_D)) = 0.$$

In addition, recall that any transition variable is distributed according to any chosen cumulative distribution $\alpha(t)$. Therefore, we can simply set our generative process to the same distribution, which ensures that:

$$\mathrm{KL}(q(\tau_1,\ldots,\tau_D)|p(\tau_1,\ldots,\tau_D)) = 0.$$

At this point we observe that the sampling times only determine the order in which the dimensions $\boldsymbol{x}$ are modelled. In fact when modelling $\boldsymbol{x}(\tau_i)|\boldsymbol{x}(\tau_{i+1})$ only one dimension dimension changes from $\tau_{i+1}$ to $\tau_i$. Since all $\{\tau_i\}$ are independently and equally distributed, the distribution over the order of $\{\tau_i\}$ is uniform. A subtle detail is that the *reverse* of the order of $\tau_i$ describes the generative order since we model timestep $\tau_i$ given $\tau_{i+1}$. Nevertheless, since the distribution over orders is uniform, the distribution over *reverse* orders is also uniform. Therefore:

$$\mathbb{E}_{q(\tau_1,\ldots,\tau_D)} \sum_i \log p(\boldsymbol{x}(\tau_i)|\boldsymbol{x}(\tau_{i+1})) = \mathbb{E}_{\sigma \sim \mathcal{U}(S_d)} \sum_i \log p(\mathbf{x}_{\sigma(i)}|\boldsymbol{x}_{\sigma(<i)}),$$

where the latter equation contains the same omitted notation as in the main paper: The model is aware which dimensions are conditioned on and which are not. In practical terms, this means that $\boldsymbol{x}_{\sigma(<i)}$ should be viewed as a masked vector and not as an order-less set. For the curious reader, technically the ability to move from order-less to the structured masked vector is enabled by conditioning on $\sigma$. In summary, modelling the generative process of a continuous absorbing jump process is equivalent to an AO-ARDM. This is beneficial, as viewing the model as an AO-ARDM gives a simple bound to the number of required steps and allows an easier perspective on the model and its implementation.

# D    EXPERIMENTAL DETAILS

In this section further details are given on the experimental setup.

**Images**    For CIFAR10 (Krizhevsky et al., 2009) we train the model using a fixed number of steps using the typical splits and evaluate the test log-likelihood after 3000 epochs of training. The results that are reported with standard deviations results are based on runs with three different initial seeds. The other results are based on single-run experiments. The runs take approximately 2 weeks to complete training on 8 TPUv4 devices, although good performance ($\approx 2.8$ bits per dimension) is already achieved after a couple of days.

As a base architecture for the OA-ARDM and the Upscale ARDMs, the exact same U-Net architecture as in (Kingma et al., 2021) are used. This architecture has 32 ResBlocks at resolution $32 \times 32$, a single middle attention layer and then another 32 ResBlocks at resolution $32 \times 32$. Throughout the network feature maps have 256 channels. This architecture is typical for NLL optimization and lossless compression, which typically require many high-resolution feature maps (Mentzer et al., 2019). The models are trained for 3000 epochs with Adam using a learning rate of 0.0001 and beta parameters (0.9 / 0.999). The input processing uses a combination of the floating point representation and an embedding layer. The integer-valued input is put through a simple normalization by dividing by the total number of classes, and subtracting 0.5. To this normalized input the mask $m$ is then concatenated. In the case of upscale ARDMs, the current stage in one-hot representation is also converted to a mask with $S$ channels, and is also part of $m$. So in that case $m$ has $S + 1$ channels. Then a $3 \times 3$ convolutional layer maps the concatenated inputs to $3/4$ of the channels. In addition, the integers are also fed through an embedding layer to $1/4$ of the channels. These two outputs are then combined which produces the feature maps with 256 channels. This is given as an input to the U-Net architecture as described above. Following Austin et al. (2021) we include the Cross-Entropy (CE) objective $\mathcal{L}_{CE} = \mathbb{E}_{t \sim \mathcal{U}(1,...,D)} \left[ D(D - t + 1)\mathcal{L}_t \right]$ (i.e. the unnormalized likelihood components), with a small factor of 0.001. However, in an experiment without the $\mathcal{L}_{CE}$ loss included, no substantial differences in performance were found for our ARDMs. Since the likelihood of the dataset is estimated with the ARDM objective, the results are computed over multiple dataset passes to reduce the stochastic effects from sampling $t$ and $\sigma$. For evaluation the exponential moving average of the parameters is used with a momentum coefficient of 0.9999. The models are trained with a batch size of 128. The gradient is clipped at 100.

**Language**    For the text8 dataset (Mahoney, 2011)[1] we train using the typical $90 \cdot 10^6 / 5 \cdot 10^6 / 5 \cdot 10^6$ splits in characters. Because the text8 dataset is a long string of characters, there is predictive information between segments when chunked. For this reason there is a big difference between model performance in literature in the reported scores on the text8 benchmark. Some methods consider a larger context before the current sequence, which greatly improves the information available to the model and gives better log-likelihoods. The runs take approximately a week to complete on 4 TPUv4 devices.

Since we are interested in the pure modelling capacity of models, we follow (Hoogeboom et al., 2021; Austin et al., 2021) and consider chunked text segments *without* any additional context. However, since the text8 splits are not evenly divisible by 256, we slightly adjust the chunk size to 250 characters, to avoid dropping the last batch. We validated empirically with a baseline Transformer that this small change does not meaningfully change the performance with the 256 version. For reference, a baseline 12 layer Transformer attains 1.35 bpc on this problem.

As a base architecture, we use a 12 layer Transformer as used in (Austin et al., 2021). It has 768 dimensions, 12 heads, 3072 MLP dimensions. For the ARDM architectures we followed (Austin et al., 2021) and used a batch size of 512 with no dropout. For standard language model baseline, since we observed overfitting the batch size was lowered and dropout of 0.1 was added. The models are trained for $3 \cdot 10^6$ training steps. ARDMs are optimized with Adam with a learning rate of 0.0005 which has a linear warm-up for the first 5000 steps. The additional $\mathcal{L}_{CE}$ loss was included with a factor 0.0001. The gradient is clipped at 0.25. For evaluation the exponential moving average of the parameters is used with a momentum coefficient of 0.995. All models use a sinusoidal

---

[1] http://mattmahoney.net/dc/text8.zip

positional embedding. However, the OA-Transformer based on the XLNet approach (Yang et al., 2019) requires both the input and target positional embeddings to infer which permutation currently needs to be generated. In (Alcorn & Nguyen, 2021), this is handled by interleaving input and target nodes in the sequence. A downside to this approach is that is increases the sequence length by two, which increases the quadratic computational complexity of the attention layers by four. In contrast, we concatenate the input and target embeddings, which does not alter the sequence length.

**Audio**   For audio experiments we used a subset of the SC09 dataset (Warden, 2018) obtained by filtering out all non-digit commands from the dataset without changing the structure of the train/-validation/test splits. The resulting dataset contains $31158/3643/4107$ training/validation/test audio clips that are 1 second long and sampled at 16 kHz. In a few rare cases when the audio clips were shorter than 1 second, we right-padded them with zeros; and all considered models were trained and evaluated on padded data. A Tensorflow Datasets TFD version of this dataset is provided with the open-source code. Training takes approximately 4 days.

For both, the AO-ARDM as well as the Upscale ARDM experiments, we closely followed the Dif-fWave setup (Kong et al., 2021) in terms of the network architecture and size, bit adapted the input embedding layer to take input masks into account. Specifically, we used a non-causal WaveNet (van den Oord et al., 2016a) architecture with 36 blocks, 256 channels and a dilation cycle of 11 (i.e. maximum dilation of $2048$); input embedding were obtained by concatenating *1)* 64-channel embeddings of integer input values; with *2)* 192-channel mask and continuous input value embeddings output by a width 3 convolution; the shared time embedding was obtained by mapping the standard 256-channel sine and cosine representation through two dense layers with $1024$ features and Swish nonlinearities (Elfwing et al., 2018).

Audio AO-ARDM and Upscale ARDM models were trained using the Adam optimizer (Kingma & Ba, 2014) with beta paramters 0.9 / 0.999 for $10^6$ steps with a batch size of 256 and a linear learning rate warm-up over the first $15000$ steps followed by a constant learning rate of $10^{-4}$. During training we tracked an exponential moving average (EMA) of the model parameters using a momentum of 0.995, and employed the EMA parameters during evaluation. As in the case of image ARDMs, the models were optimized using a combination of the ELBO and CE objectives - the latter taken with a tiny weight of $10^{-4}$. No further regularisation was used.

Due to the large output space ($2^{16}$ classes) audio AO-ARDM modelled the output distribution using a mixture of discretized logistics (DMoL) with 30 components, although we experimentally found the number of components to not make a big difference. To aid with training, the DMoL was initialized as an approximately uniform distribution with different mixtures responsible for the different parts of this distribution; and gradients with an $L_2$ norm larger than $1000$ were re-normalized. Owing to the smaller per-stage output space, we were able to utilize the categorical softmax parameterization for the Upscale ARDMs. Empirically we observed this model class to demonstrate a more stable training behaviour (in a addition to significantly improved likelihoods), which we (partially) attribute to the choice of parametrization.

For our autoregressive single-order baseline we sought to deviate from the above AO-ARDM setup as little as possible, and used a causal version of the WaveNet architecture above. However, we observed that the single-order baseline overfits quickly on the training data. To overcome this, we found it necessary to use weight decay (0.01), smaller batch size (64) and fewer channels (128) for the baseline model.

