# OpenReview forum: "Autoregressive Diffusion Models"
_ICLR.cc/2022/Conference — ICLR 2022 Poster_

### Official Review · Reviewer_Pog3 · 2021-10-20

**Correctness:** 4
**Technical Novelty And Significance:** 3
**Empirical Novelty And Significance:** 2
**Recommendation:** 6
**Confidence:** 3

**Main Review:**

Strengths:
+ Intuitive and novel approach that aims to combine benefits from two different modeling techniques
+ Extension of a widely used model class (autoregressive models)
+ Vast practical applicability

Weaknesses:
- Performance gains in benchmarking tests appear to be rather moderate
- The number of benchmarking tests is limited; robustness of the result remains unclear



**Summary Of The Paper:**

This work introduces a new model class combining elements from autoregressive and discrete diffusion models. The new approach is named Autoregressive Diffusion Models (ARDMs). The authors show good performance compared to alternative generative models, in particular in lossless image compression. The authors demonstrate that the new model has particular benefits compared to the autoregressive and discrete diffusion models in terms implementation efficiency, scalability and parallelization.

**Summary Of The Review:**

Overall, this is a well reported work that presents useful new modeling ideas that bring together previously separate methodology.  The work could, however, benefit from a more comprehensive set of benchmarking with different data sets / case studies to establish how robust the claimed performance gains are.

---

> ### Author Response · Authors · 2021-11-11
> **Response to reviewer Pog3**
>
> The reviewer mentions two weaknesses of the work, which we address below.
>
> Firstly the reviewer mentions that performance gains seem to be moderate. This really depends on which experiment one looks at: it is true that for text8 the performance difference between the OA-ARDM and the D3PM-absorbing with 1000 steps is small. However, as we explain in the paper, this is expected since the 1000 step model is approaching a continuous time limit. The point here is not the difference in bpd, but the improved performance in that only 250 steps are required to approach this limit with ARDMs. For a comparable number of steps, the performance difference is more pronounced (e.g. 1.43 vs 1.47 for ~250 steps and 1.51 vs 1.56 for 20 steps).
>
>
> Secondly, the reviewer mentions that the number of benchmarking tests is limited, and that the robustness of the results remains unclear.
> We believe that the wide variety of different benchmarks (on text, audio and images) demonstrates the wide applicability of ARDMs and their ability to achieve good performance on these domains. Other reviewers agree that the empirical evaluation is extensive. In terms of robustness, we would like to point out that for well-performing models Table 1 and 2 contain standard deviations estimated over three runs with different initial seeds. As can be seen in those tables, the models are quite robust. For example, the deviations between runs is only 0.001 for the OA-ARDM on text8 and 0.005 for the OA-ARDM on CIFAR10.
>
> We hope that this response has clarified the reviewer’s concerns.

---

> > ### Comment · Reviewer_Pog3 · 2021-11-15
> > **Responses ok; no changes to score**
> >
> > The responses are adequate and well aligned with the other review comments. No changes to the original score.

---

### Official Review · Reviewer_Y5ay · 2021-10-27

**Correctness:** 4
**Technical Novelty And Significance:** 2
**Empirical Novelty And Significance:** 2
**Recommendation:** 5
**Confidence:** 3

**Main Review:**

Let me start by confessing that I really struggled with this paper, because (in my humble opinion) it is written with the assumption for the reader to be familiar with not yet published (or recently, where recently means days, accepted) work. In principle, I would be fine with this, if the submitted paper was self-contained such that it would be possible to have a grasp on the presented ideas, to be refined upon reading the literature.
Unfortunately, at least for me, this was not the case: I imperatively had to gain familiarity with unpublished work to fully appreciate the technical merit of this paper. Specifically, this is true for discrete diffusion processes (heavily based on a paper recently accepted at NeurIPS 2021, and a paper updated on Arxiv as recently as few days ago) and for the dynamic programming part (an arxiv paper appeared a few months ago).

As another disclaimer (which instead is not affecting my positive judgment for the paper), I found the experimental section to require extensive knowledge in "application" domains where I have passing familiarity (language modeling, neural compression) but by no means I consider myself as an expert.

With all that being said, here is my detailed review:

* Section 1:
One suggestion I have to improve the intro is to focus more on the key issues (especially computational ones) that are addressed by this work. Also, upscaling seems to be the most original contribution of this work, and it is only mentioned in passing. I would instead avoid to go into technicalities that can be known only by substantial familiarity with unpublished or recently accepted work, such as Austin et al. 2021.
(By the way, I really liked the paper Austin et al. 2021; this is a necessary read prior to working on this paper!)

* Section 2:
The background is not enough, and all the accent is put on order agnostic methods. This is, in my humble opinion, problematic because there is no mention to discrete diffusion models, which are necessary to understand this work.
Another note: with reference to Yang et al. 2019, it is brought to the reader attention that if you shuffle the order of the r.v., you can still have a reasonable bound to the log likelihood, but the training procedure requires model architectures that are invariant to the permutation order, and this looks like a limitation you want to avoid. This is important in my opinion, and not sufficiently exploited or emphasized, e.g. in the experimental section.

* Section 3:
I think here I need more references and a more thorough discussion about the requirement for triangular or causal dependence to parametrize an autoregressive model, which seems to be the one pain point you aim to address. Also, I think that the goal of upscaling variables comes out of the blue, and it is not until later in the paper that we learn about the modeling approach to see it as a diffusion process, that we learn the connection with the presented work.

Fig. 1 and 2 are very useful to understand the key idea of the proposed parametrization! One thing that induced me in error the first (of the many) time I read this paper is the use of $\theta$: I generally consider these are parameters, whereas this is the output of a "neural network".

Overall, I really miss a clear connection to discrete diffusion processes in the first part of Section 3. Instead, the reader is assumed to be familiar with the details, and the final discussion before subsection 3.1 does not help.

* Section 3.1:
I consider this as a key contribution, even though it is based on an existing dynamic programming approach. The observation in eq (3) is key.
I was wondering, what would happen if the "shape" of $L_t$ wouldn't be so well behaved as you depict in Figure 3. Although $L_t$ is monotonically decreasing, it could be so that with a small budget you would incur in a larger error than we may think: for example, consider a linearly decreasing $L_t$.

* Section 3.2:
This is, in my opinion, the main contribution of this work, in that as I see it, it is the most original. However, it again assumes complete familiarity with an unpublished work (Austin et al., 2021), and little context and background information is given. As a consequence, it is difficult to appreciate, for example, the parametrization presented in this work (the transition matrices here it are deterministic, theirs are stochastic).

Section 5 is very well executed, I have no major comments (nor all the required expertise to nitpick on experimental setup choices).

So to conclude:

== Strength:
* methods to improve the efficiency of autoregressive models
* upscaling as a discrete autoregressive diffusion model
* compelling experimental results

== Weaknesses:
* the paper is largely not self-contained, making it hard to fully grasp its merits
* the discrete probabilistic diffusion nature of the proposed models is not put forward enough
* main methodological contribution drowned into a series of careful engineering contributions



**Summary Of The Paper:**

This is a very dense paper that proposes a new autoregressive modeling (ARM) approach, which enjoys several properties. The proposed approach builds on two main ideas: 1) order-agnostic ARM which they exploit to randomize over the generation of variables according to a random order picked uniformly from all possible permutations of orders, 2) denoising diffusion probabilistic models extended to work on discrete variables.
Then, one contribution of this paper is to improve the computational cost of order-agnostic ARM. This stems from rewriting the log likelihood component at a given timestep $t$, and noticing that this single term can be optimized for all datapoints in a minibatch simultaneously. The parametrization of the neural network proposed in this work is also an improvement over a traditional approach to generation in an autoregressive diffusion model.
Furthermore, eq (3) is key for the realization that the un-ordered generation of $k$ tokens can happen independently, because this would contribute to $k$ times the log likelihood at a given timestep $t$. Using ideas from a related work, in this paper the authors show how to exploit this fact and a dynamic programming approach to parallelize autoregressive diffusion models for a given budget.

An additional contribution is to consider the application of the proposed model to the problem of upscaling. This is modeled as a (discrete) diffusion process where the forward process is destructive (downscaling) and the backward process is constructive (upscaling).
As discussed originally in the work used by the authors as a basis for discrete diffusion processes, upscaling can be defined using simple transition matrices, which define the transitions in the Markov chain supporting the process from a downscaled version of the data to an upscaled version. The authors present two variants of a parametrization approach to learn the backward distributions.

An extensive experimental campaign complements the ideas presented in this paper, with application ranging from the image to the text domains, including upscaling and lossless compression.

**Summary Of The Review:**

It is not an easy task (for me) to come up with a recommendation for this paper. I didn't find any technical flaws, I think the topic is very interesting and important, and the empirical validation of the proposed method(s) is extensive.

Nevertheless, I find this paper has problems with the positioning, and with the choice of what the authors consider their main contribution. Also, a lot of the proposed methods are based on other (recent) work, which are assumed to be perfectly known by the reader: this makes this paper not self-contained, and to some extent reduces the surface of the contributions, which seem additional engineering prowesses on top of existing methods.

---

> ### Author Response · Authors · 2021-11-11
> **Response to reviewer Y5ay**
>
> Firstly the reviewer mentions that it is difficult to clearly understand all the connections in this paper to literature, especially the ones first released as recently as a couple of months ago. Further, the reviewer suggests putting discrete diffusion in the background section.
>
> - We agree with the reviewer. To clarify the connection with diffusion, we have added a background section on (absorbing) discrete diffusion and added a clarification in section 3 about the similarity between OA-ARDMs and absorbing diffusion. In section 3.2 we have also added a clarification of the connection to diffusion processes. Note that in addition to the changes described above, the original paper already contained the connection to diffusion in the Related work (section 4) and by extension in Appendix C.
>
> Further, the reviewer suggests adding a more thorough discussion on triangular dependence in section 3 and motivating upscaling better, and adding reference to discrete diffusion earlier.
>
> - In the updated paper we have extended our description of the difficulty of enforcing triangular dependence. which we support with two references.  In addition, we will draw the connection to diffusion in section 3.2 a bit earlier so that the goal is clearer from the start.
>
>
> And finally, the reviewer raises two more questions. The reviewer asks: In section 3.1, what would happen if Lt is monotonically decreasing in a different way, for instance a linear decreasing Lt?
> - If the shape of Lt would be linear, then the algorithm would find the optimal solution which would be an equally spaced grid (i.e. a linspace). Indeed as the reviewer points out, in this case perhaps more steps are required for reasonable performance. However, in practice we find for all data modalities (images, text, audio) that the shape of the graph of Lt is curved.
>
>
> And secondly in section 3.2. The reviewer mentions that this can be seen as the main contribution of this work. But it requires more context knowledge from Austin et al. to understand this part better.
> - It is always difficult to balance trying to avoid re-explaining a method that is publicly available with giving sufficient context to understand the method. To aid the reader, we have described the connection with diffusion for this specific part. In addition the background has been updated, so that all in all the connection is clearer. Furthermore, although we are happy that the reviewer views section 3.2 as an important contribution, we think that the reviewer may have overlooked subtle other contributions in earlier sections. For instance, it is not only this section that is connected to absorbing diffusion. Rather, it is already the OA-ARDM from earlier in section 3 that is the continuous-time limit of absorbing diffusion. This connection is explained in detail in section 4 and appendix C.
>
> We hope that this response has clarified the reviewer’s concerns.

---

### Official Review · Reviewer_g6Qq · 2021-11-02

**Correctness:** 4
**Technical Novelty And Significance:** 4
**Empirical Novelty And Significance:** 4
**Recommendation:** 8
**Confidence:** 4

**Main Review:**

## 1. Strengths

- a. Strong contributions:
     - ARDM generalize a wide class of models (autoregressive models, order agnostic autoregressive models (XL-Net))
     - ARDM allow unfolding of the generative process with *depth upscaling* (translates in using more denoising steps)
     - ARDM allow parallelizing the generative process for fast generation
- b. even the authors introduce multiple contributions, the paper is well written, easy to read and well-structured.
- c. Equations stated in the main text are correct
- d. Related work is well described. This is very clear how the method relates to D3PM.
- e. Ideas are supported by a complete set of experiments
    - The idea translates into impressive empirical results
    - The experiments are well designed and interpreted, this gives a good understanding of the functioning of the OA-ARDM
    - Additional results in the appendix give a good picture of the underlying generative process

## 2. Weaknesses
- a. Training time and required resources are not mentioned.
- b. Experiments do not describe how models trained with a single seed are selected (does the method consistently converge to the reported results, or have you selected the best performing seed?).
- c. The code needs to be released for full reproducibility (which is planned according to the authors)

## 3. Minor comments / questions to the authors
- a. Could you please elaborate on the limitations of the dynamic programming approach for parallelisation?
- b. Is the method sensitive to the choice of hyperparameters?

## 4.Disclaimer
- I am confident in my evaluation of the definition of the OA-ARDM, overall terms of parallelisation, depth upscaling and modelling
- I only have a general understanding of the described dynamic program and limited experience with the compression experiments

**Summary Of The Paper:**

Autoregressive Diffusion Models (ARDM) are discrete diffusion models that extend D3PMs and generalize order agnostic auto-regressive models (OA-ARM) . Similarly to other diffusion models, training OA-ARDMs only require evaluating a single transition step (i.e $p(x_{\sigma(t:t+k)} | x_{\sigma(<t}))$, where $\sigma$ is a random permutation). In contrast to traditional ARMs, OA-ARMs are trained to generate k steps $p(x_{\sigma(t:t+k)})$ at each transition.

Furthermore, the authors supplement the OA-ARM with two additional contributions:
- factoring the generative model across the variable *depth* (named *depth upscaling*), which allows unfolding the generative process into an even greater number of steps ($S \times T$ steps instead of $T$)
- an algorithm to parallelize generation using dynamic programming (allows trading quality for generation speed)

The authors demonstrate the effectiveness of OA-ARM on character-level text modelling (text8), image modelling/compression (CIFAR-10).

**Summary Of The Review:**

This paper is well-written and introduces significant contributions. The contributions are theoretically significant (generalization of ARMs) and are of practical use (parallelization of the OA-ARDMs, depth upscaling). The method is well related to the literature. The theoretical findings are supported by a complete set of experiments.

---

> ### Author Response · Authors · 2021-11-11
> **Response to reviewer g6Qq**
>
> The reviewer mentions the following weaknesses:
>
> a. Training time and required resources are not mentioned.
> We agree with the reviewer that these can be useful and have included these in Appendix D.
>
> b. Experiments do not describe how models trained with a single seed are selected.
> We want to emphasize that in Table 1 & 2 multiple seed runs are reported for the well-performing models. Here one can see that there is very little deviation between runs: e.g 0.001 for the OA-ARDM on text8 and 0.005 for the OA-ARDM on CIFAR10.
>
> c. The code needs to be released for full reproducibility.
> In the meantime we have released the code publicly. To avoid breaking anonymity we will not share the link here.
>
> The reviewer has two minor further questions.
> 1) The reviewer asks about the limitations of the DP approach. The DP approach has complexity O(n^3) where n the total number of timesteps, although several components of the algorithm are parallelized (e.g.. the argmin call). For this reason, processes with very large time steps may be expensive to solve and would need to be approximated with a coarser measurement grid first. In practice we are able to run processes with 16000 steps in about 25 seconds and 3072 in under two seconds. We will extend appendix A.2 with this information.
> 2) The reviewer asks about sensitivity to hyperparameters: From a handful of experiments testing different optimization parameters, the method did not seem to be particularly sensitive to hyperparameters, with models reaching comparable performance.
>
> We hope that this response has clarified the reviewer’s concerns.
>
> Edit: Improved formatting slightly

---

> > ### Comment · Reviewer_g6Qq · 2021-11-16
> > **Thanks for the improvements, I am more confident in recommending your paper**
> >
> > Thank you for your feedback and for answering the weak points that I have mentioned. I consider these weak points answered and solved.
> >
> > The DP approach remains the part that I am the least confident in reviewing. I hope other reviewers can be more reliable judges. Nonetheless, I have been through the numpy/jax code provided in appendix A.2. I now have a better understanding. A visualisation of how the triangular matrix is used at every step could help the least experimented readers.
> >
> > Although I agree with other reviewers that reading your work was time demanding, as it required reviewing related papers, I believe this is ok to rely on existing literature, as long as you precisely describe how this applies to your work.
> >
> > I remain confident in recommending this paper.

---

### Official Review · Reviewer_y93W · 2021-11-02

**Correctness:** 4
**Technical Novelty And Significance:** 3
**Empirical Novelty And Significance:** 4
**Recommendation:** 6
**Confidence:** 3

**Main Review:**

Strength
- The paper is well-written.
- Various experiments are conducted (across diverse domains), and hence, it shows the extensibility of their work.
- The experimental results are very promising in two ways: performance measure, and efficiency on the number of steps.
- (Minor) The authors explicitly provide the limitation of the their work.

Weakness
- The implementation is not provided (even though the authors provide algorithms and pseudo-codes, and promised to release the code).
- No error bars are provided in the experimental result.
- Question: does each step takes similar time? Else, time consumption might be provided.

**Summary Of The Paper:**

The paper proposes Autoregressive Diffusion Models (ARDMs), which is a combination of two concepts: autoregressive models and (discrete) diffusion models. The proposed ARDMs generalize order-agnostic ARMs [1] and discrete diffusion models [2]. ARDMs are efficient in several ways: they do not require the causal dependency so that the can be trained in efficient way, and they provide the parallel generation. The authors provide proper experiments that back up their work.

[1] Uria et al., A deep and tractable density estimator, 2014.
[2] Austin et al., Structured denoising diffusion models in discrete state-spaces, 2021.

**Summary Of The Review:**

I lean to positive on this work. The work generalizes two frameworks: OA-ARM and discrete diffusion model. The provided method is simple to implement and efficient is generating mode. Also, the authors conduct experiments across various domains, and the result are very promising. Therefore, I recommend this paper to be accepted in ICLR 2022.

---

> ### Author Response · Authors · 2021-11-11
> **Response to reviewer y93W**
>
> The reviewer raises three points:
>
> Firstly, the reviewer mentions that the implementation of our code is not provided. We are happy to say that in the meantime we have already released our code and it is publicly available. To preserve anonymity however, we will not share the link here.
>
> Secondly, the reviewer mentions that there is no indication of performance difference between runs. It is true that we did not provide standard deviations for every run, because there are so many experiments and it would be expensive to run all models multiple times. However, for well-performing models in Table 1 & 2 the standard deviations over multiple initial seeds are given. Note that the difference between runs in performance is small, with deviations of 0.001 for the OA-ARDM on text8 and 0.005 for the OA-ARDM on CIFAR10.
>
> Finally, the reviewer asks if each step takes a similar amount of time. The answer is yes, each step uses exactly a single forward pass. When models use the same architecture, results are directly comparable (as is the case in Table 1).
>
> We hope that this response has clarified the reviewer’s concerns.

---

> > ### Comment · Reviewer_y93W · 2021-11-29
> > **Thank you for the response.**
> >
> > Thank you for the response and the clarification. For the meantime, I keep my score.

---

### Decision · Program_Chairs · 2022-01-20

**Decision:**

Accept (Poster)

**Comment:**

This paper introduces Autoregressive Diffusion Models (ARDMs), which generalises order-agnostic autoregressive models and absorbing discrete diffusion.

All reviewers appreciated the paper with a few also finding it very dense. The experimental section is a bit lacking in detail. This has to some degree been answered in the discussion and should also be included in the final version of the paper.

Acceptance is recommended.